# Reasoning on the Manifold: Bidirectional Consistency for Self-Verification in Diffusion Language Models

**Jiaoyang Ruan** [* 1] **Xin Gao** [* 1 2] **Yinda Chen** [3] **Hengyu Zeng** [1] **Liang Du** [4] **Guanghao Li** [1] **Jie Fu** [2] **Jian Pu** [1]

## Abstract

While Diffusion Large Language Models (dLLMs) offer structural advantages for global planning, efficiently verifying that they arrive at correct answers via valid reasoning traces remains a critical challenge. In this work, we propose a geometric perspective: Reasoning on the Manifold. We hypothesize that valid generation trajectories reside as stable attractors on the high-density manifold of the learned distribution, whereas invalid paths exhibit off-manifold drift. To operationalize this, we introduce Bidirectional Manifold Consistency (BMC), a training-free, unsupervised metric that quantifies the stability of the generated sequence through a forward-masking and backward-reconstruction cycle. Empirically, we demonstrate BMC's versatility across the full reasoning lifecycle: (1) in Diagnosis, it serves as a robust discriminator of solution validity without ground truth answer; (2) in Inference, it enables rejection resampling to effectively concentrate computational resources on complex reasoning tasks; and (3) in Alignment, it functions as a dense geometric reward that transforms sparse outcome supervision into fine-grained guidance, empowering models to self-evolve beyond standard baselines. Our results establish intrinsic geometric stability as a robust indicator of correctness for dLLMs.

## 1. Introduction

Large Language Models (LLMs) have fundamentally transformed natural language processing, yet the dominant au-

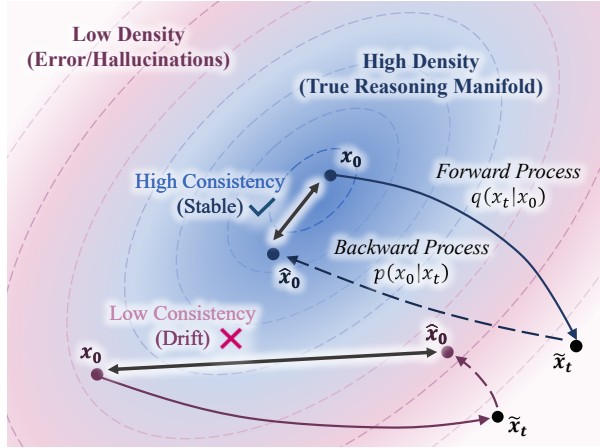

*Figure 1.* **Geometric Intuition of BMC.** BMC evaluates the validity of $x_0$ by probing its stability under a forward-backward cycle. Valid solutions (blue) function as stable attractors on the high-density manifold, enabling faithful reconstruction ($\hat{x}_0 \approx x_0$). Conversely, erroneous outputs (purple) exhibit off-manifold drift, causing the reconstruction to diverge significantly.

toregressive (AR) paradigm remains constrained by its strict left-to-right order. This sequential dependency limits the ability to perform global planning or revise earlier decisions during complex tasks. Recently, diffusion Large Language Models (dLLMs) have emerged as a compelling non-AR alternative (Li et al., 2025; Yu et al., 2025) that offers structural advantages suited for reasoning. Unlike the causal masking of AR models, dLLMs employ full attention to perceive the entire context (Nie et al., 2025; Ye et al., 2025), which enables the global planning of reasoning structures. Moreover, their generative dynamics are governed by a bidirectional process of noise injection and denoising (Austin et al., 2021). This temporal structure facilitates the iterative deliberation characteristic of System 2 reasoning, enabling the model to revisit tokens for fine-grained control and global optimization (Zhao et al., 2025).

Despite these structural capabilities, dLLMs face a critical challenge shared by the broader landscape of generative AI: the verification of generation reliability. As models are increasingly deployed for complex tasks, the risk of hallucinations and error accumulation within reasoning traces demands rigorous verification (Zhang et al., 2025b; Ji et al.,

---
[*]Equal contribution [1]Institute of Science and Technology for Brain-Inspired Intelligence, Fudan University, Shanghai, China [2]Shanghai Artificial Intelligence Laboratory, Shanghai, China [3]University of Science and Technology of China, Hefei, China [4]IEG, Tencent Inc., Shenzhen, China. Correspondence to: Jian Pu and Jie Fu <jianpu@fudan.edu.cn, jiefu@pjlab.org>.

*Proceedings of the 43rd International Conference on Machine Learning*, Seoul, South Korea. PMLR 306, 2026. Copyright 2026 by the author(s).

2023; Gao et al., 2026). Current verification methods, primarily developed for the AR paradigm, typically rely on extrinsic signals that treat the model as a black box. For instance, Process Reward Models (PRMs) (Zheng et al., 2025; Wang et al., 2024; Lightman et al., 2023) require expensive human annotation to train external discriminators, while sampling-based techniques like self-consistency (Chen et al., 2023; Brown et al., 2024; Wang et al., 2022) incur high computational overhead and often falter on hard samples where models systematically converge to incorrect consensus. Crucially, these generic strategies fail to exploit the unique probabilistic dynamics of diffusion models. This gap motivates a shift from external supervision to internal discovery: *Does the dLLM generation process itself contain intrinsic signals correlated with solution validity?*

We answer affirmatively by proposing a geometric perspective: *Reasoning on the Manifold*, which hypothesizes that valid solutions reside on the high-density manifold as stable attractors, whereas erroneous sequences exhibit detectable off-manifold drift. As illustrated in Figure 1, this geometric distinction dictates reconstruction stability: valid solutions recover faithfully after perturbation. Conversely, erroneous outputs undergo semantic drift, characterized by high inconsistency where the reconstructed trajectory diverges significantly from the original generation. To quantify this, we introduce an unsupervised metric **Bidirectional Manifold Consistency (BMC)**. Theoretically, BMC approximates log-likelihood by measuring local geometric stability. Crucially, this evaluation requires only a few denoising steps for reconstruction, allowing BMC to serve as a robust, training-free indicator of solution validity with minimal overhead.

To our knowledge, this is the first work to systematically exploit the intrinsic dynamics of dLLMs for self-verification. We validate BMC as a unified framework: (1) *Diagnosis*: BMC demonstrates superior discriminative power in error detection across four reasoning benchmarks, significantly outperforming consistency baselines; (2) *Inference*: We introduce Manifold-Guided Iterative Resampling (MGIR), which uses BMC to effectively concentrate computational resources on complex queries; and (3) *Alignment*: In Reinforcement Learning (RL), BMC augments sparse outcome rewards into dense guidance signals, enabling models to self-evolve towards higher logical consistency.

Our main contributions are summarized as follows:

- We propose Reasoning on the Manifold, a geometric perspective that anchors logical validity to the high-density regions of the learned distribution, treating correctness as an intrinsic manifold constraint.
- We introduce Bidirectional Manifold Consistency (BMC), an unsupervised metric that leverages the reversible dynamics of dLLMs to quantify generation correctness.
- We validate BMC across the full reasoning lifecycle,

demonstrating its versatility in error diagnosis, inference-time self-correction, and RL alignment.

## 2. Related Work

**Diffusion Large Language Models.** dLLMs have emerged as a compelling non-autoregressive alternative to standard LLMs (Austin et al., 2021; Hoogeboom et al., 2021). Unlike sequential AR generation, dLLMs leverage global context via all-to-all attention (Nie et al., 2025), effectively modeling generation as trajectory evolution on a learned data manifold (Song et al., 2020). Recent scaling efforts, such as LLaDA (Nie et al., 2025) and Dream (Ye et al., 2025), have demonstrated competitive performance on large-scale benchmarks. Notably, recent frameworks have harnessed this iterative nature for dynamic self-correction: RemeDi (Huang et al., 2025b) employs a dual-stream mechanism to identify and re-mask low-confidence tokens, while CDLM (Zhang et al., 2025a) explicitly trains models to detect and rectify errors through corrective post-training. While these approaches leverage diffusion dynamics for generative refinement, our work first formalizes implicit geometric properties into an explicit verification criterion.

**Verification of Reasoning.** Verifying reasoning chains is essential for enabling System 2 search capabilities. Existing approaches fall into three categories. First, discriminative methods like PRMs (Lightman et al., 2023; Wang et al., 2024) provide granular supervision but require costly annotations or expensive rollouts. Second, generative verifiers (Zhang et al., 2024) leverage model reasoning to produce critiques, yet suffer from self-correction blind spots (Tsui, 2025), failing to detect errors without external ground truth. Third, execution-based frameworks like Loong (Huang et al., 2025a) scale verification using code interpreters but depend on 12 domain-specific sandboxes, limiting applicability in open-ended tasks. BMC addresses these limitations by exploiting the white-box geometry of dLLMs to quantify solution stability via forward-backward cycles.

**Self-Correction and Reward Modeling.** The intrinsic self-correction capability of LLMs remains a contentious topic; Huang et al. (2023) argue that models struggle to rectify errors in the absence of external feedback. Although adaptive sampling techniques (Kumar et al., 2024) offer partial mitigation, they often necessitate task-specific retraining. In the domain of alignment, RLHF (Ouyang et al., 2022) and GRPO (Shao et al., 2024) depend heavily on human preferences or ground-truth oracles. Self-rewarding frameworks (Yuan et al., 2024) attempt to close this loop but are ultimately bottlenecked by the model's subjective biases. While recent dLLM-specific methods like TraceRL (Wang et al., 2025b) enhance consistency by optimizing trajectory likelihood approximations over d1 (Zhao et al., 2025), they operate implicitly rather than providing an explicit metric

for trajectory consistency. BMC bridges these gaps via a unified geometric signal that enables training-free correction and serves as a dense reward for self-alignment.

# 3. Theoretical Analysis

In this section, we provide the preliminaries and theoretical analysis. We begin by defining BMC for dLLMs (§3.1). We then establish its equivalence to log-likelihood maximization (§3.2) and characterize it geometrically as a measure of stability during manifold projection (§3.3).

## 3.1. Preliminaries and Definition

We consider the problem of verification over discrete sequences. Let $x_0 = [x_0^{(1)}, \ldots, x_0^{(L)}]$ be a sequence of tokens where each $x_0^{(i)}$ belongs to a finite vocabulary $\mathcal{V}$. To model the generation process, we adopt the Masked dLLMs with notations following Austin et al. (2021):

**Forward Diffusion Process.** We define a forward process that progressively corrupts $x_0$ into a sequence of purely absorbing states. Let $m \notin \mathcal{V}$ denote a special [MASK] token. The transition probability at timestep $t$ is defined by the matrix $Q_t = (1 - \beta_t)I + \beta_t \mathbb{1} e_m^\top$, where $\beta_t$ controls the noise schedule. Let $\alpha_t = 1 - \beta_t$ and $\bar{\alpha}_t = \prod_{s=1}^{t} \alpha_s$ denote the cumulative signal schedule. Applying this transition over time yields a factorized marginal $q(x_t|x_0) = \prod_i q(x_t^{(i)}|x_0^{(i)})$, where each token is independently retained or masked:

$$q(x_t^{(i)}|x_0^{(i)}) = \bar{\alpha}_t \, \mathbb{I}(x_t^{(i)} = x_0^{(i)}) + (1 - \bar{\alpha}_t) \, \mathbb{I}(x_t^{(i)} = m). \tag{1}$$

This formulation implies that at any step $t$, the sequence $x_t$ acts as a partial observation of $x_0$, retaining original tokens with probability $\bar{\alpha}_t$ while masking the rest.

**Reverse Denoising Process.** To generate samples, we utilize the reverse process $p(x_{t-1}|x_t)$. We adopt the $x_0$-parameterization, where the model $f_\theta(x_t)$ is trained to predict the clean tokens $x_0$ directly. In the absorbing state setting, this objective is structurally equivalent to Masked Language Modeling (MLM). The reverse step is analytically derived using the posterior rule by marginalizing the tractable forward posterior $q(x_{t-1}|x_t, x_0)$ over the model's predicted distribution $p_\theta(\tilde{x}_0|x_t)$:

$$p_\theta(x_{t-1}|x_t) \propto \sum_{\hat{x}_0 \in \mathcal{V}} q(x_{t-1}|x_t, \hat{x}_0) p_\theta(\hat{x}_0|x_t). \tag{2}$$

The model is trained by minimizing a variational lower bound combined with an auxiliary cross-entropy loss. For absorbing state diffusion, only masked tokens contribute to the objective, which simplifies to:

$$\mathcal{L}_{\text{D3PM}}(\theta) = \mathbb{E}_{t,x_0,\tilde{x}_t}[-\sum_{i \in M_t} \log p_\theta(x_0^{(i)}|\tilde{x}_t)], \tag{3}$$

where $\tilde{x}_t \sim q(x_t|x_0)$ is the corrupted state and $M_t = \{i \mid \tilde{x}_t^{(i)} = \text{[MASK]}\}$ denotes the masked indices. This objective upper-bounds the negative log-likelihood, encouraging recovery of the original tokens from corrupted states.

Based on this bidirectional mechanism, we quantify the validity of a generated sequence by measuring its reconstruction discrepancy under the diffusion process. This metric assesses how well the model can recover the original sequence $x_0$ when subjected to masking perturbations:

**Definition 3.1** (Bidirectional Manifold Consistency (BMC)). Given a generated sequence $x_0$, let $\tilde{x}_t$ be the corrupted state obtained by re-masking under the forward process $q(x_t|x_0)$ at timestep $t$. Let $\hat{x}_0(\tilde{x}_t)$ denote the reconstruction derived using the denoiser $f_\theta$. The BMC score $\mathcal{R}_\mathcal{D}(x_0)$ is defined as the expected reconstruction similarity:

$$\mathcal{R}_\mathcal{D}(x_0) := -\mathbb{E}_{t \sim \mathcal{U}(1,T), \tilde{x}_t} [\mathcal{D}[x_0, \hat{x}_0(\tilde{x}_t)]], \tag{4}$$

where $\mathcal{D}(\cdot, \cdot)$ is a general dissimilarity measure quantifying the discrepancy between original $x_0$ and the reconstruction.

## 3.2. Theoretical Connection to Likelihood

To validate the theoretical correctness of BMC, we first analyze its behavior when $\mathcal{D}$ is instantiated as the Kullback-Leibler (KL) divergence, establishing a formal connection to the Evidence Lower Bound (ELBO).

**Proposition 3.2** (BMC as Reweighted ELBO Estimator). *Let $\mathcal{D}$ in Eq. 3.1 be the KL divergence between the Dirac distribution of the clean data $\delta_{x_0}$ and the model reconstruction $P_\theta(\cdot \mid \tilde{x}_t)$. The BMC score $\mathcal{R}_{\text{KL}}(x_0)$ constitutes an estimator of the Reweighted ELBO:*

$$\mathcal{R}_{KL}(x_0) = \frac{1}{T} \sum_{t=1}^{T} \mathbb{E}_{q(\tilde{x}_t|x_0)} [\log p_\theta(x_0|\tilde{x}_t)] + C, \tag{5}$$

*where $T$ is the total number of diffusion steps and $C$ represents constants independent of $\theta$.*

*Proof Sketch.* The KL divergence reduces to cross-entropy for Dirac distributions. The reweighted ELBO is derived by Austin et al. (2021). See Appendix A.1 for details. $\square$

While Proposition 3.2 grounds BMC in probability theory via KL divergence, strict likelihood implies lexical rigidity, failing to distinguish paraphrases from logical errors. To reconcile verification with both *decision correctness* and *semantic diversity*, we instantiate the generic measure $\mathcal{D}$ (Definition 3.1) with generalized metrics over either discrete or continuous space. We now prove that these metrics remain valid proxies for likelihood optimization: discrete metrics maintain consistency with $\mathcal{R}_{\text{KL}}(x_0)$, while continuous metrics provide a relaxation for synonymy.

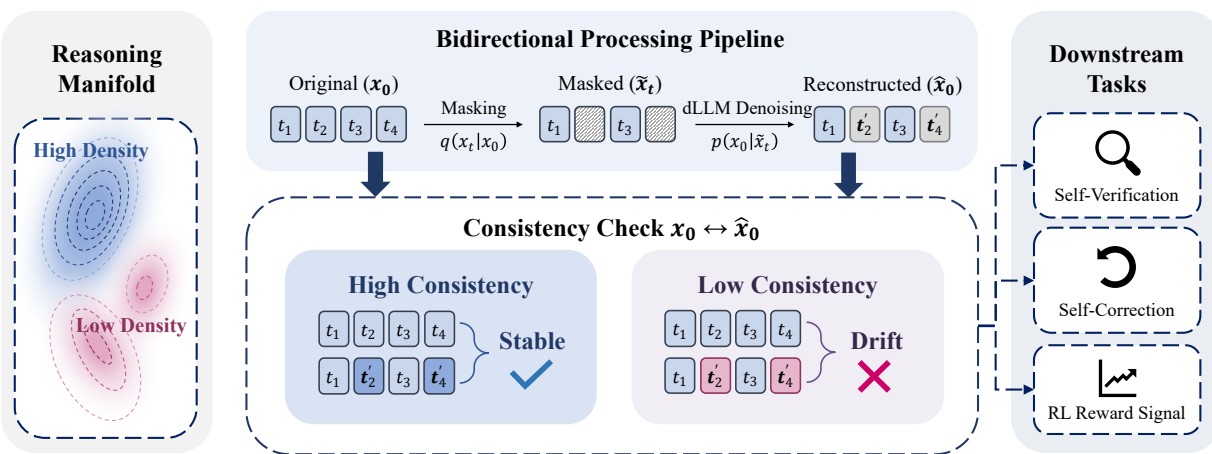

*Figure 2.* **The BMC Framework. Left:** Correct solutions (blue) occupy stable high-density regions on the reasoning manifold; incorrect solutions (purple) lie off-manifold. **Center:** Bidirectional pipeline: masking $x_0 \to x_t$, reconstruction $x_t \to \hat{x}_0$, consistency check. High consistency indicates stability; drift reveals errors. **Right:** Downstream applications—verification, correction, and RL alignment.

**Proposition 3.3** (Consistency with Marginal Reweighted ELBO). *Let $\mathcal{I} \subseteq \{1, \ldots, L\}$ be the set of indices corresponding to decision-critical tokens in the original sequence $x_0$. Define $\mathcal{D}_f$ as a Csiszár $f$-divergence generated by a strictly convex function $f$ with $f(1) = 0$. For each token $x_0^{(i)}$ in the critical set $\mathcal{I}$, let $\hat{x}_0^{(i)}(\tilde{x}_t)$ be the corresponding token in the reconstruction $\hat{x}_0(\tilde{x}_t)$. Then, the BMC score $\mathcal{R}_{\mathcal{D}}(x_0)$, defined as the negative expected divergence, is equivalent to maximizing the Marginal Reweighted ELBO of the critical subsequence $x_0^{(\mathcal{I})}$.*

*Proof Sketch.* Using strict convexity of function $f$. See Appendix A.2 for full derivation. □

**Proposition 3.4** (Semantic Continuity of Likelihood). *Let $\Phi : \mathcal{V}^L \to \mathbb{R}^d$ be a sequence embedding function that maps token sequences to a continuous embedding space, and let $d_{\Phi}(x, y)$ be a dissimilarity measure induced by $\Phi$ between sequences $x$ and $y$ in this space. Assume that the conditional log-likelihood $\log p_\theta(x \mid \tilde{x}_t)$ is locally $K$-Lipschitz continuous with respect to $d_{\Phi}(x, y)$. For any original sequence $x_0$ and its reconstruction $\hat{x}_0$ within a local neighborhood, if their semantic discrepancy is bounded by $d_{\Phi}(x_0, \hat{x}_0) \le \epsilon$, then the likelihood ratio is constrained as follows:*

$$e^{-K\epsilon} \le \frac{p_\theta(\hat{x}_0 \mid \tilde{x}_t)}{p_\theta(x_0 \mid \tilde{x}_t)} \le e^{K\epsilon}. \quad (6)$$

*Proof Sketch.* Using local $K$-Lipschitz continuity of the log-likelihood. Full derivation in Appendix A.3. □

These propositions provide theoretical guarantees for BMC, ensuring the robustness of the consistency score under our general definition. In the token space, we can use a generalized $f$-divergence to measure the matching of critical tokens, such as verifying key reasoning steps or exact conclusions,

which corresponds to maximizing the Reweighted ELBO and thus log-likelihood. In the embedding space, we can relax semantic constraints by using metrics like L2 norm or cosine similarity to measure the similarity between the original and reconstructed sequence embeddings, enabling semantic flexibility (e.g., paraphrasing) while maintaining close alignment with the original log-likelihood.

### 3.3. Geometric Interpretation

While the probabilistic perspective aligns BMC with the likelihood of a sequence, it does not fully address the calibration gap, where fluent yet factually incorrect outputs receive high probability. To address this, we analyze the geometric stability of the generated sequence by examining its reconstruction error in the continuous latent space.

Let $\mathcal{Z} \subset \mathbb{R}^{L \times d}$ be the continuous embedding space of generated sequences. We define the *Denoising Operator* $\mathcal{T}_\theta : \mathcal{Z} \to \mathcal{Z}$ as the expected one-step reconstruction:

$$\mathcal{T}_\theta(z) := \mathbb{E}_{\tilde{z} \sim q(\cdot|z)} \left[ \mathbb{E}_{z' \sim p_\theta(\cdot|\tilde{z})}[z'] \right]. \quad (7)$$

$\mathcal{T}_\theta$ captures the deterministic mean field flow of the diffusion dynamics. We posit that the manifold of valid solutions, $\mathcal{M} \subset \mathcal{Z}$, acts as a set of stable fixed-point attractors (where $\mathcal{T}_\theta(z^*) = z^*$ for any $z^* \in \mathcal{M}$). Accordingly, we model the trained denoiser $\mathcal{T}_\theta$ as a local contraction mapping that projects off-manifold states toward validity.

**Proposition 3.5** (Manifold Distance Upper Bound). *Let $\mathcal{Z}$ be equipped with a metric induced by the embedding norm. Assume $\mathcal{T}_\theta$ is a $\kappa$-contraction mapping locally around $\mathcal{M}$ with rate $0 \le \kappa < 1$. For any generated sequence $z_0$, let $z^* \in \mathcal{M}$ be the closest valid solution. The geometric error is upper-bounded by the reconstruction residual:*

$$\|z_0 - z^*\| \le \frac{1}{1 - \kappa} \|z_0 - \mathcal{T}_\theta(z_0)\|. \quad (8)$$

*Proof Sketch.* Applying the triangle inequality and the fixed-point property. Detailed derivation in Appendix A.4. □

## 4. Method

As the exact computation of the theoretical consistency score $\mathcal{R}_\mathcal{D}(x_0)$ formulated in Section 3 is computationally intractable, we propose an unsupervised, training-free BMC estimator that efficiently leverages the intrinsic bidirectional dynamics of dLLMs. As illustrated in Figure 2, we first implement a perturbation-recovery cycle and then operationalize the abstract measure $\mathcal{D}$ using concrete similarity metrics. This yields a practical stability signal for error diagnosis, inference guidance, and alignment.

### 4.1. The BMC Estimator

The estimation begins with the *Forward Perturbation*. We utilize the intrinsic forward transition of the discrete diffusion model to project the complete generated sequence $x_0$ onto a partially observable state $\tilde{x}_t$. Let $m \in \{0,1\}^L$ be a binary mask vector sampled from a Bernoulli distribution with parameter $1 - \gamma$, where $\gamma$ is the masking ratio. The perturbed state $\tilde{x}_t$ is obtained by replacing tokens with the absorbing token [MASK] where $m_i = 0$:

$$\tilde{x}_t^{(i)} = m_i x_0^{(i)} + (1 - m_i) \,[\text{MASK}]. \qquad (9)$$

This operation forces the model to reproduce the original trajectory from a fragmented state.

Subsequently, we perform *Backward Reconstruction* using the pre-trained denoiser $p_\theta$. Starting from the absorbing state $\tilde{x}_t$, we execute $K$ iterative denoising steps with a linear schedule, to generate a reconstructed trajectory $\hat{x}_0 \sim p_\theta(\cdot|\tilde{x}_t; K)$. A critical advantage of this estimator is its computational efficiency. Unlike the generative process which requires the full trajectory length $T$, our verification step performs only a truncated reconstruction with $K \ll T$ (e.g., $K = 16$ vs. $T = 1024$). This allows BMC to probe the manifold stability of a solution with a fraction of the cost required for resampling.

While the perturb-and-reconstruct protocol itself is general, the dissimilarity measure $\mathcal{D}(x_0, \hat{x}_0)$ must be instantiated to match the decision-critical structure of the target domain, and alternative instantiations are discussed in Appendix D.3. Specifically, we leverage the fact that maximizing the negative dissimilarity is equivalent to maximizing the similarity between the original and reconstructed states. To provide a comprehensive measurement of stability, we compute a composite score $S_{\text{BMC}}$ derived from six metrics widely adopted for evaluating generation fidelity. Let $M = \{i \mid m_i = 0\}$ be the set of masked indices. The components are defined as follows:

*Token Accuracy* ($s_{\text{tok}}$) monitors local convergence by mea-

**Algorithm 1** Manifold-Guided Rejection Sampling

**Require:** Query $q$, Model $p_\theta$, Threshold $\tau$, Budget $N_{\max}$, Mask Rate $\gamma$, Steps $K$
**Ensure:** Reliable solution sequence $x^*$
1: $x_{\text{best}} \leftarrow$ None, $\quad S_{\text{best}} \leftarrow -1$
2: **for** $n = 1$ to $N_{\max}$ **do**
3: $\quad$ **Generate:** $x_0 \sim p_\theta(\cdot|q)$
4: $\quad$ **Perturb:** $\tilde{x}_t \sim q(x_t|x_0)$ with rate $\gamma$ {Eq. (9)}
5: $\quad$ **Truncated Reconstruct:** $\hat{x}_0 \sim p_\theta(\cdot|\tilde{x}_t; K)$
6: $\quad$ **Verify:** $S \leftarrow S_{\text{BMC}}(x_0, \hat{x}_0)$
7: $\quad$ **if** $S > \tau$ **then**
8: $\quad\quad$ **return** $x_0$ {Early exit: Stable solution found}
9: $\quad$ **end if**
10: $\quad$ **if** $S > S_{\text{best}}$ **then**
11: $\quad\quad$ $S_{\text{best}} \leftarrow S$, $\quad x_{\text{best}} \leftarrow x_0$
12: $\quad$ **end if**
13: **end for**
14: **return** $x_{\text{best}}$ {Fallback to most stable candidate}

suring the exact reconstruction rate of masked tokens:

$$s_{\text{tok}} = \frac{1}{|M|} \sum_{i \in M} \mathbb{I}(x_0^{(i)} = \hat{x}_0^{(i)}). \qquad (10)$$

*Semantic Similarity* ($s_{\text{sem}}$) accounts for valid paraphrasing by computing the cosine similarity between sentence embeddings $\phi(x_0)$ and $\phi(\hat{x}_0)$:

$$s_{\text{sem}} = \frac{\phi(x_0)^\top \phi(\hat{x}_0)}{\|\phi(x_0)\|\|\phi(\hat{x}_0)\|}. \qquad (11)$$

*Number Retention* ($s_{\text{num}}$) captures the stability of critical logic nodes in mathematical reasoning. Let $\mathcal{N}(x_0)$ be the multiset of numbers extracted from sequence $x_0$. We define stability as the recall rate:

$$s_{\text{num}} = \frac{|\mathcal{N}(x_0) \cap \mathcal{N}(\hat{x}_0)|}{|\mathcal{N}(x_0)| + \epsilon}. \qquad (12)$$

*Final Answer Match* ($s_{\text{ans}}$) captures the terminal convergence of the reasoning trajectory. Using a task-specific extractor $E(\cdot)$, we define a binary indicator:

$$s_{\text{ans}} = \mathbb{I}(E(x_0) \equiv E(\hat{x}_0)). \qquad (13)$$

*Auxiliary Metrics*: We additionally track *Character Similarity* ($s_{\text{char}}$), the normalized Levenshtein ratio, to ensure morphological robustness; and *Intrinsic Confidence* ($s_{\text{conf}}$), the average predicted probability during reconstruction, to verify if the trajectory lies in a high-density region.

The final BMC score is a weighted linear combination $S_{\text{BMC}}(x_0) = \sum_k \lambda_k s_k$, where weights $\lambda_k$ are aggregation coefficients that balance different consistency aspects.

### 4.2. Manifold-Guided Inference

The BMC estimator $\mathcal{S}_{\text{BMC}}(x_0)$ serves as a versatile geometric signal for inference, quantifying the structural reliability of a generated sequence. In the context of complex reasoning, the generation landscape is dominated by plausible but fallacious hallucinations, while true solution paths function as distinct *stable attractors* within the learned distribution.

Standard decoding methods often struggle to distinguish these rigorous paths from locally fluent but logically drifting errors, leading to confident hallucinations.

To address this, we propose *Manifold-Guided Rejection Sampling (MGRS)*, detailed in Algorithm 1, which transforms generation into an adaptive stability search. Using $\mathcal{S}_{\mathrm{BMC}}$ to quantify geometric stability, we establish a dynamic acceptance criterion: candidates exceeding a threshold ($\mathcal{S}_{\mathrm{BMC}} > \tau$) are identified as stable solutions residing on the learned manifold. These are accepted immediately, enabling the efficient resolution of simpler queries. Conversely, low-stability outputs trigger an iterative resampling loop, directing computational resources toward exploring alternative trajectories until a geometrically consistent solution emerges. This approach effectively focuses compute on hard queries, leveraging intrinsic stability to distinguish robust trajectory from unstable drift.

### 4.3. Geometric Alignment via Dense Rewards

While inference-time selection optimizes existing trajectories, we seek to internalize the model manifold by shaping the policy $\pi_\theta$ to favor stable reasoning paths. We achieve this by incorporating the BMC score as a dense RL reward. To mitigate the risk of reinforcing consistent but erroneous chains, we propose a gated reward function that conditions geometric stability on logical validity:

$$r(x_0) = \mathbb{I}(y_{\mathrm{pred}} = y^*) \cdot [r_{\mathrm{base}} + \alpha_t \cdot S_{\mathrm{BMC}}(x_0)], \quad (14)$$

where $\mathbb{I}(\cdot)$ is the correctness indicator for the final answer and $r_{\mathrm{base}}$ is a constant completion reward.

This multiplicative formulation establishes a strict hierarchy where validity serves as a prerequisite for stability assessment. Consequently, incorrect responses receive zero reward regardless of their internal consistency, preventing the optimization of high-confidence errors. For correct responses, the term $\alpha_t \cdot S_{\mathrm{BMC}}$ provides a fine-grained gradient that distinguishes between unstable spurious success and robust reasoning anchored on the learned manifold.

To balance solution discovery and trajectory refinement, we implement a progressive curriculum for $\alpha_t$. Since strong early constraints can prematurely restrict exploration, we adopt a linear annealing schedule: $\alpha_t = \alpha_{\min} + (\alpha_{\max} - \alpha_{\min}) \cdot \frac{t}{T}$, where $t$ is the current step and $T$ is the total budget. This prioritizes answer correctness in the early stages ($\alpha_t \approx \alpha_{\min}$) while progressively emphasizing intrinsic geometric stability as training converges ($\alpha_t \to \alpha_{\max}$).

For policy optimization, we adopt the Sandwiched Policy Gradient (SPG) framework (Wang et al., 2025a). Unlike standard diffu-GRPO which relies on biased ELBO approximations for negative advantages (Zhao et al., 2025), SPG leverages sandwiched evidence bounds to accurately estimate gradients for intractable dLLM likelihoods.

## 5. Experiments

We evaluate BMC across the reasoning lifecycle via three progressive objectives: (1) Unsupervised Error Diagnosis, assessing the estimator's discriminative power in identifying erroneous trajectories; (2) Adaptive Self-Correction, leveraging BMC for manifold-guided resampling to enhance inference-time performance and sample efficiency; and (3) Geometric Alignment, employing BMC as a dense reward signal to guide RL for policy refinement.

### 5.1. Experimental Setup

**Models and Datasets.** We implement BMC on two state-of-the-art dLLMs: LLaDA-8B-Instruct (Nie et al., 2025) and Dream-v0-Instruct-7B (Ye et al., 2025). To benchmark reasoning performance, we utilize four datasets with varying complexity: GSM8K (Cobbe et al., 2021) for grade-school math, MATH (Hendrycks et al., 2021) for high-school competition problems, ARC-Challenge (Clark et al., 2018) for knowledge-intensive scientific reasoning, and GPQA (Rein et al., 2024) for expert-level science questions.

**Baselines and Metrics.** For error diagnosis, we evaluate Model Confidence (Kadavath et al., 2022) and Self-Evaluation (Madaan et al., 2023), using AUROC (Bradley, 1997) and AUPR (Davis & Goadrich, 2006) to measure discrimination. For self-correction, we benchmark against Standard Sampling, Best-of-$N$ (Stiennon et al., 2020), and Self-Consistency (Wang et al., 2022), reporting Pass@1 Accuracy and Sample Efficiency, which is defined as the accuracy gain per additional sample. For alignment, we compare with SFT (Zhao et al., 2025) and Outcome RL (Wang et al., 2025a). See implementation details in Appendix C.1.

**Implementation Details.** BMC uses masking ratio $\gamma = 0.9$ and $K = 16$. Semantic similarity employs all-MiniLM-L6-v2 (Reimers & Gurevych, 2019); intrinsic confidence averages masked probabilities. MGRS sets $\tau = 0.75$ and budget $N_{\max} = 10$. RL alignment uses $r_{\mathrm{base}} = 1.5, \alpha \in [0.5, 1.0]$. Full hyperparameters in Appendix C.3.

### 5.2. Discriminative Performance in Error Diagnosis

We assess the discriminative capacity of BMC to distinguish erroneous trajectories from correct solutions without ground-truth. Table 1 reports the AUROC and AUPR scores across four benchmarks of increasing complexity.

Conventional baselines exhibit significant degradation as task complexity increases. Likelihood-based metrics, such as Model Confidence, and prompting-based methods like Self-Evaluation, yield AUROC scores near 0.5 on ARC-C and GPQA, essentially reducing to random guessing. In con-

*Table 1.* **Unsupervised Error Diagnosis Performance.** We report the AUROC and AUPR scores for LLaDA-8B-Instruct (Nie et al., 2025) and Dream-v0-Instruct-7B (Ye et al., 2025) across four reasoning benchmarks.

| Model | Method | GSM8K | | MATH | | ARC-C | | GPQA | |
|---|---|---|---|---|---|---|---|---|---|
| | | AUROC↑ | AUPR↑ | AUROC↑ | AUPR↑ | AUROC↑ | AUPR↑ | AUROC↑ | AUPR↑ |
| LLaDA | Model Confidence | 0.753 | 0.879 | 0.713 | 0.430 | 0.550 | 0.824 | 0.482 | 0.256 |
| | Self-Consistency | 0.872 | 0.929 | 0.803 | 0.546 | 0.735 | 0.895 | 0.539 | 0.281 |
| | Self-Evaluation | 0.549 | 0.737 | 0.558 | 0.266 | 0.546 | 0.821 | 0.529 | 0.257 |
| | **BMC (Ours)** | **0.893** | **0.943** | **0.820** | **0.613** | **0.777** | **0.918** | **0.678** | **0.375** |
| Dream | Model Confidence | 0.641 | 0.468 | 0.522 | 0.228 | 0.463 | 0.855 | 0.468 | 0.290 |
| | Self-Consistency | 0.684 | 0.498 | 0.675 | 0.408 | 0.708 | 0.922 | 0.527 | 0.313 |
| | Self-Evaluation | 0.565 | 0.401 | 0.525 | 0.245 | 0.570 | 0.886 | 0.539 | 0.332 |
| | **BMC (Ours)** | **0.898** | **0.787** | **0.825** | **0.660** | **0.804** | **0.951** | **0.605** | 0.329 |

*Table 2.* **Adaptive Self-Correction Performance.** We compare the MGRS dynamic sampling strategy against fixed-budget baselines ($N = 3$). "Eff" denotes Sample Efficiency: $\text{Eff} = (\text{Acc} - \text{Acc}_{\text{std}})/(N_{\text{avg}} - 1)$.

| Model | Method | GSM8K | | MATH | | ARC-C | | GPQA | |
|---|---|---|---|---|---|---|---|---|---|
| | | Acc↑ | Eff↑ | Acc↑ | Eff↑ | Acc↑ | Eff↑ | Acc↑ | Eff↑ |
| LLaDA | Standard Sampling | 70.5 | - | 24.4 | - | 83.2 | - | 24.8 | - |
| | Self-Consistency | 74.3 | 1.86 | 24.2 | -0.10 | 86.1 | 1.45 | 25.0 | 0.11 |
| | Best-of-N (Confidence) | 70.7 | 0.08 | 23.8 | -0.30 | 83.3 | 0.04 | **27.7** | **1.45** |
| | Best-of-N (BMC) | 77.8 | 3.64 | 26.0 | **0.80** | 86.3 | 1.54 | 25.0 | 0.11 |
| | **MGRS (Ours)** | **79.5** | **3.98** | **27.6** | 0.66 | **87.2** | **1.85** | 26.1 | 0.49 |
| Dream | Standard Sampling | 72.0 | - | 22.4 | - | 74.7 | - | 22.3 | - |
| | Self-Consistency | 72.4 | 0.19 | 20.2 | -1.10 | 80.6 | 2.91 | 29.5 | 3.57 |
| | Best-of-N (Confidence) | 72.5 | 0.23 | 20.2 | -1.10 | 80.4 | 2.82 | 29.5 | 3.57 |
| | Best-of-N (BMC) | 72.5 | 0.23 | 20.2 | -1.10 | 80.5 | 2.86 | **29.7** | **3.68** |
| | **MGRS (Ours)** | **72.9** | **0.69** | **22.6** | **0.05** | **81.0** | **3.21** | 27.9 | 1.20 |

trast, BMC maintains discriminative validity across all domains (e.g., 0.678 AUROC on GPQA with LLaDA), demonstrating that geometric stability captures structural signals imperceptible to simple token probabilities.

While Self-Consistency (SC) is a competitive baseline, BMC demonstrates superior robustness where consensus assumptions falter. Specifically, BMC's advantage over SC widens as task difficulty increases and correct solutions become sparse. On LLaDA, this gap grows from 2.1% on GSM8K to 13.9% on expert-level GPQA, validating that intrinsic stability is more reliable than population statistics. On Dream-7B, SC underperforms due to limited diversity and repetitive errors. BMC mitigates this by evaluating manifold consistency rather than redundancy, yielding 0.898 AUROC on GSM8K (+21.4%) and 0.605 on GPQA (+7.8%). These results confirm that BMC is robust to variations in base model accuracy and sampling diversity.

### 5.3. Efficiency of Adaptive Self-Correction

We assess the performance of MGRS iterative resampling compared to fixed-budget baselines. As shown in Table 2 and Table 6, our adaptive strategy consistently achieves superior accuracy-efficiency trade-offs.

On LLaDA, MGRS secures the highest accuracy across GSM8K (79.5%), MATH (27.6%), and ARC (87.2%). Notably, this is achieved with minimal computational overhead;

for instance, on GSM8K, the method requires only 3.3 samples on average, yielding a sample efficiency of 3.98. This contrasts sharply with Confidence-based Best-of-N, which often degrades performance (e.g., negative efficiency on MATH), confirming that token probability is a poor proxy for reasoning validity compared to geometric stability.

A critical advantage of the proposed method is its ability to align computational budget with task difficulty. The average sample count naturally scales from simpler tasks (GSM8K: ~2.2–3.3) to complex reasoning (MATH: ~5.4–5.8). This geometric awareness enables the model to accept stable solutions early while reserving compute for unstable trajectories. Although Best-of-N (BMC) marginally outperforms the guided approach on GPQA with Dream-7B (29.7 vs 27.9), the guided strategy remains competitive while strictly adhering to the manifold stability criterion.

### 5.4. Effectiveness of Geometric Alignment

We design a composite reward that augments sparse outcome rewards with BMC as a dense incentive. We evaluate our method, denoted as Geometric Alignment, against two primary baselines: fine-tuning (SFT) and standard RL optimized exclusively for answer correctness (Outcome RL). To provide broader context, we also include state-of-the-art AR models in Table 3, positioning dLLM reasoning capabilities against established AR benchmarks.

*Table 3.* **Accuracy (%) on Reasoning Tasks.** We compare our Geometric Alignment against baseline SFT, Outcome RL, and AR models. For dLLMs, we report results across generation lengths (128, 256, 512 tokens). † indicates results from published papers.

| Model/Method | GSM8K | | | MATH | | | ARC-C | | | GPQA | | |
|---|---|---|---|---|---|---|---|---|---|---|---|---|
| *AR Models* | | | | | | | | | | | | |
| LLaMA3-8B† | | 53.1 | | | 18.4 | | | 82.4 | | | 25.9 | |
| Gemma2-9B† | | 76.7 | | | 44.3 | | | 68.4 | | | 32.8 | |
| Qwen2.5-7B† | | 85.4 | | | 41.1 | | | 57.5 | | | 36.4 | |
| *Gen Length* | 128 | 256 | 512 | 128 | 256 | 512 | 128 | 256 | 512 | 128 | 256 | 512 |
| *dLLM (LLaDA)* | | | | | | | | | | | | |
| + SFT | 69.4 | 77.9 | 80.4 | 26.8 | 32.8 | 34.8 | 83.7 | 83.2 | 78.1 | 25.0 | 27.0 | 26.8 |
| + Outcome RL | 76.5 | 82.9 | 83.5 | 32.6 | 35.8 | 37.2 | 83.1 | 83.5 | 82.2 | 36.2 | **35.0** | 30.8 |
| + **Geometric Align (Ours)** | **78.3** | **85.4** | **85.8** | **36.6** | **40.2** | **41.6** | **85.5** | **85.9** | **85.2** | **36.4** | 33.5 | **34.4** |

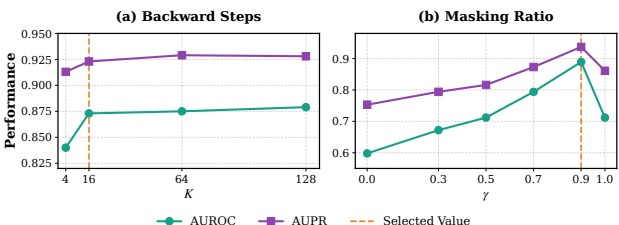

*Figure 3.* **Hyperparameter Sensitivity of the Bidirectional Process on GSM8K.** (a) Backward Process: Reconstruction steps $K$. (b) Forward Process: Perturbation masking ratio $\gamma$ in Eq. 9.

*Table 4.* **Ablation of BMC Components.** We evaluate the discriminative power (AUROC/AUPR) of individual similarity metrics against the composite BMC score on GSM8K.

| Metric Configuration | AUROC | AUPR |
|---|---|---|
| *Individual Metrics* | | |
| Cross-Entropy ($s_{ce}$) | 0.576 | 0.764 |
| Token Accuracy ($s_{tok}$) | 0.640 | 0.783 |
| Semantic Similarity ($s_{sem}$) | 0.659 | 0.796 |
| Intrinsic Confidence ($s_{conf}$) | 0.699 | 0.817 |
| Number Retention ($s_{num}$) | 0.796 | 0.889 |
| Character Similarity ($s_{char}$) | 0.638 | 0.785 |
| Final Answer Match ($s_{ans}$) | 0.819 | 0.916 |
| *Composite Strategies* | | |
| Surface-Level ($s_{tok} + s_{sem} + s_{conf}$) | 0.684 | 0.807 |
| Surface + Logic ($+s_{num}$) | 0.753 | 0.850 |
| Composite + Cross-Entropy | 0.794 | 0.874 |
| Composite (Uniform Weights) | 0.821 | 0.901 |
| **Composite (Optimized Weights)** | **0.880** | **0.927** |

On reasoning-intensive tasks, our method consistently outperforms Outcome RL across all generation lengths (e.g., +4.4% on MATH at 512 tokens). While standard Outcome RL treats all correct answers identically, potentially reinforcing spurious chains that accidentally arrive at the solution, BMC modulates the reward signal to prioritize geometrically stable trajectories. This dense guidance effectively mitigates reward hacking by filtering out unstable hallucinations, allowing the model to match strong AR baselines (e.g., Qwen2.5-7B) using intrinsic stability signals.

On knowledge-heavy benchmarks, performance is sensitive to generation length, as extended chains can introduce noise in direct retrieval tasks. However, our method demonstrates superior robustness, preventing the degradation often observed in unconstrained generation (85.3% for ARC-C and 34.4% for GPQA with 512 generation length).

### 5.5. Ablation Studies

We examine the impact of key hyperparameters and metric components to validate the key design of BMC.

**Hyperparameter Configuration.** Figure 3 reveals distinct geometric behaviors underlying BMC's design choices. Performance improves sharply with backward diffusion steps, saturating at $K = 16$ ($0.840 \rightarrow 0.873$ AUROC), which confirms that truncated reconstruction suffices to probe the local contraction properties of the denoising operator $\mathcal{T}_\theta$. The masking ratio exhibits a critical inverted-U trend peaking at $\gamma = 0.9$ (AUROC 0.889). Notably, the sharp drop

at full masking ($\gamma = 1.0$: 0.712 AUROC) validates that BMC requires residual local context as geometric anchors to test the stability of the original reasoning path, distinguishing it from unconstrained generative resampling. We use $N_{BMC} = 4$ ensemble samples. Detailed ablations including computational costs are in Appendix D.2.

**Consistency Metrics.** Table 4 dissects BMC's components. Among individual features, Final Answer Match (0.819) and Number Retention (0.796) perform best, confirming that key semantic elements matter more than surface-level matching (0.640). In contrast, Cross-Entropy ($s_{ce}$) yields the lowest AUROC (0.576); while theoretically grounded, its token-wise sensitivity to non-critical tokens introduces excessive noise. Since its inclusion degrades composite performance (0.821 to 0.794), we exclude $s_{ce}$ from the final score to prioritize structural stability. Combining multiple features with uniform weights achieves 0.821 AUROC. Task-specific optimized weights further improve the AUROC to 0.880.

## 6. Conclusion

This work introduces Bidirectional Manifold Consistency (BMC), a training-free framework leveraging the intrinsic bidirectional dynamics of dLLMs for self-verification. By

characterizing solution validity as geometric stability, BMC resolves standard likelihood miscalibration. Empirically, this proxy achieves up to 14% AUROC gains in error diagnosis and 4–8× efficiency improvements in adaptive self-correction. Furthermore, incorporating BMC as a dense RL reward enables models to internalize manifold structures and autonomously self-evolve. By systematically exploiting these dynamics, our findings provide a principled geometric foundation for verifying and aligning dLLMs.

## Impact Statement

This paper introduces Bidirectional Manifold Consistency, also known as BMC, as a novel framework for self verification in diffusion based language models. By grounding reasoning validity in the geometric stability of the learned manifold, our work provides a mathematically principled alternative to opaque black box scoring methods. The broader impact of this research lies in enhancing the reliability of autonomous reasoning systems because BMC enables models to detect hallucinated logic that may appear linguistically fluent but is structurally inconsistent. This capability is crucial for deploying AI in high stakes domains such as legal analysis, scientific discovery, and education where ensuring the integrity of the reasoning process is as important as the correctness of the final output. Furthermore, as a training free metric, BMC promotes more energy efficient AI alignment by reducing the dependency on massive, human annotated datasets for reinforcement learning.

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

# Appendix

# A. Detailed Proofs

**Introductory Note:** The propositions below establish conditional motivating results under standard structural assumptions, providing geometric intuition for BMC rather than strict formal guarantees for deployed models.

## A.1. BMC as Reweighted ELBO Estimator (Proposition 3.2)

In this section, we provide a rigorous derivation connecting the proposed Bidirectional Manifold Consistency (BMC) score to the Evidence Lower Bound (ELBO). We show that maximizing the BMC score is mathematically equivalent to maximizing a reweighted variational lower bound, which serves as a standard surrogate objective in diffusion model training.

*Proof.* Let $x_0$ denote the clean data and $\tilde{x}_t$ the corrupted state at timestep $t$, sampled from the forward process $q(\tilde{x}_t|x_0)$.

We first analyze the KL divergence term in the BMC definition for a fixed pair $(x_0, \tilde{x}_t)$. Since the ground truth $x_0$ is deterministic, we model its distribution as a Dirac delta $\delta_{x_0}$ (or equivalently, a one-hot distribution in discrete space). The KL divergence then expands as:

$$
\begin{aligned}
\mathcal{D}_{\text{KL}}(\delta_{x_0}||p_\theta(\cdot|\tilde{x}_t)) &= \sum_x \delta(x - x_0) \log \frac{\delta(x - x_0)}{p_\theta(x|\tilde{x}_t)} \\
&= \underbrace{\sum_x \delta(x - x_0) \log \delta(x - x_0)}_{\text{Entropy of } x_0 \text{ (Constant } C_1)} - \sum_x \delta(x - x_0) \log p_\theta(x|\tilde{x}_t) \\
&= -\log p_\theta(x_0|\tilde{x}_t) + C_1.
\end{aligned}
\tag{15}
$$

Note that the entropy term vanishes for discrete deterministic data. The BMC score $\mathcal{R}_{\text{KL}}(x_0)$ is defined as the negative expectation of this divergence over diffusion timesteps $t \sim \mathcal{U}(1, T)$ and noise realizations $\tilde{x}_t \sim q(\tilde{x}_t \mid x_0)$:

$$
\begin{aligned}
\mathcal{R}_{\text{KL}}(x_0) &:= -\mathbb{E}_{t,\tilde{x}_t} \left[ \mathcal{D}_{\text{KL}}(\delta_{x_0}||p_\theta(\cdot|\tilde{x}_t)) \right] \\
&= -\mathbb{E}_{t,\tilde{x}_t} \left[ -\log p_\theta(x_0|\tilde{x}_t) + C_1 \right] \\
&= \mathbb{E}_{t\sim\mathcal{U}(1,T),\tilde{x}_t\sim q(x_t|x_0)} \left[ \log p_\theta(x_0|\tilde{x}_t) \right] + C',
\end{aligned}
\tag{16}
$$

where $C'$ absorbs all entropy constants that are independent of the model parameters $\theta$.

Next, we establish the connection to the variational lower bound. It is well known that the data log-likelihood admits a lower bound via the ELBO. For discrete diffusion models with absorbing states, Austin et al. (2021) showed that the standard ELBO decomposes into a weighted sum of reconstruction log-probabilities:

$$
\log p(x_0) \geq \mathcal{L}_{\text{ELBO}}(x_0) = \sum_{t=1}^{T} \frac{1}{t} \cdot \mathbb{E}_{q(\tilde{x}_t|x_0)} \left[ \log p_\theta(x_0|\tilde{x}_t) \right] + C,
\tag{17}
$$

where the $1/t$ weighting arises from the specific noise schedule of the absorbing process.

However, in practice, training diffusion models with the strict VLB weighting often leads to suboptimal performance. As shown in Ho et al. (2020) and Austin et al. (2021), a reweighted objective that treats all timesteps uniformly has been empirically demonstrated to better balance global structure and local detail. We denote this reweighted objective as:

$$
\mathcal{L}_{\text{reweight}}(x_0) := \sum_{t=1}^{T} \mathbb{E}_{q(\tilde{x}_t|x_0)} \left[ \log p_\theta(x_0|\tilde{x}_t) \right].
\tag{18}
$$

Comparing Eq. (16) with Eq. (18), we observe that the BMC score corresponds precisely to the expectation form of this reweighted objective. Since the log-likelihood $p_\theta(x_0|\tilde{x}_t)$ factorizes over positions, and only the masked tokens contribute to the reconstruction objective, we can decompose the sequence-level log-probability into a sum over the masked index set $M_t = \{i \mid \tilde{x}_t^{(i)} = [\text{MASK}]\}$. Using the definition of expectation over a discrete uniform distribution, $\mathbb{E}_{t\sim\mathcal{U}(1,T)}[\cdot] = \frac{1}{T} \sum_{t=1}^{T}[\cdot]$, we obtain a direct linear relationship:

$$
\mathcal{R}_{\text{KL}}(x_0) = \frac{1}{T} \sum_{t=1}^{T} \mathbb{E}_{q(\tilde{x}_t|x_0)} \left[ \sum_{i \in M_t} \log p_\theta(x_0^{(i)}|\tilde{x}_t) \right] + \text{const.}
\tag{19}
$$

Consequently, maximizing the BMC score is mathematically equivalent to maximizing the reweighted ELBO. Since $\mathcal{L}_{\text{reweight}}$ serves as a high-fidelity proxy for the data log-likelihood in state-of-the-art discrete diffusion models, BMC provides a theoretically grounded metric for evaluating generative consistency. $\qquad\square$

### A.2. Consistency with Marginal Reweighted ELBO (Proposition 3.3)

*Proof.* Let the verification metric on the critical subset be defined as the negative expected divergence:

$$\mathcal{S}_{BMC} = -\mathbb{E}_{t,\tilde{x}_t}\left[\sum_{i\in\mathcal{I}}\mathcal{D}(x_0^{(i)}, \hat{x}_0^{(i)})\right]. \tag{20}$$

For each token $i \in \mathcal{I}$, let $P = \delta_{x_0^{(i)}}$ denote the one-hot distribution of the ground truth token, and $Q = p_\theta(\cdot|\tilde{x}_t)^{(i)}$ the model's predicted distribution at position $i$. The sub-metric minimizes a Csiszár $f$-divergence, which takes the form:

$$\mathcal{D}_f(P\|Q) = \sum_{v\in\mathcal{V}} q(v)f\left(\frac{p(v)}{q(v)}\right) = f\left(\frac{1}{q(x_0^{(i)})}\right)q(x_0^{(i)}) + f(0)(1 - q(x_0^{(i)})). \tag{21}$$

Let $y = q(x_0^{(i)})$ denote the predicted probability of the correct token. Since $f$ is strictly convex with $f(1) = 0$, we have $\frac{\partial \mathcal{D}_f}{\partial y} \le 0$. Therefore, minimizing the divergence $\mathcal{D}_f$ is strictly monotonic with respect to maximizing the model's predicted probability $p_\theta(x_0^{(i)}|\tilde{x}_t)$ for the target token. Thus, we can conclude that maximizing $\mathcal{S}_{BMC}$ is equivalent to maximizing the model's predicted probability for each critical token:

$$\max \mathcal{S}_{BMC} \iff \forall i \in \mathcal{I}, \max p_\theta(x_0^{(i)}|\tilde{x}_t). \tag{22}$$

Discrete diffusion models typically employ a factorized observation model (Austin et al., 2021), where the reconstruction probability of the sequence at step $t$ is the product of independent token probabilities:

$$p_\theta(x_0|\tilde{x}_t) = \prod_{k=1}^{L} p_\theta(x_0^{(k)}|\tilde{x}_t). \tag{23}$$

By the definition of marginal probability, summing out the non-critical tokens, we have:

$$\log p_\theta(x_0^{(\mathcal{I})}|\tilde{x}_t) = \sum_{i\in\mathcal{I}} \log p_\theta(x_0^{(i)}|\tilde{x}_t). \tag{24}$$

Therefore, maximizing the aggregate BMC score (sum of monotonic functions of individual probabilities) corresponds to maximizing the expectation of the marginal distribution $p_\theta(x_0^{(\mathcal{I})}|\tilde{x}_t)$ over the critical subset. $\qquad\square$

### A.3. Semantic Continuity of Likelihood (Proposition 3.4)

*Proof.* The objective of this proof is to show that the conditional log-likelihood of a reconstructed sequence is bounded by its semantic similarity to the original sequence in the embedding space.

Let $\mathcal{Z} \subset \mathbb{R}^d$ denote the continuous embedding space where the pre-trained diffusion model learns a smooth probability density. The latent log-likelihood function is defined as:

$$f(z) = \log p_\theta(x \mid \tilde{x}_t), \tag{25}$$

where $\tilde{x}_t$ is the corrupted sequence at timestep $t$, and $x$ is the sequence represented in the embedding space.

We assume that the log-likelihood function $f(z)$ satisfies local Lipschitz continuity on the data manifold $\mathcal{M} \subset \mathcal{Z}$, meaning that for any $z_1, z_2 \in \mathcal{M}$, there exists a constant $K > 0$ such that:

$$|f(z_1) - f(z_2)| \le K\|z_1 - z_2\|. \tag{26}$$

This condition implies that the change in the log-likelihood between any two points on the manifold is bounded by their distance, scaled by $K$.

Let $z = \Phi(\hat{x}_0)$ and $z^* = \Phi(x_0)$ denote the embeddings of the generated sequence $\hat{x}_0$ and the reference sequence $x_0$, respectively. Applying the Lipschitz continuity condition to the log-likelihood function, we have:

$$|\log p_\theta(\hat{x}_0 \mid \tilde{x}_t) - \log p_\theta(x_0 \mid \tilde{x}_t)| = |f(z) - f(z^*)| \leq K\|z - z^*\|. \tag{27}$$

The difference in embeddings $\|z - z^*\|$ represents the semantic discrepancy between the original sequence $x_0$ and the reconstructed sequence $\hat{x}_0$. Assuming that this discrepancy is bounded by $\|z - z^*\| \leq \epsilon$, we obtain:

$$|\log p_\theta(\hat{x}_0 \mid \tilde{x}_t) - \log p_\theta(x_0 \mid \tilde{x}_t)| \leq K\epsilon. \tag{28}$$

Exponentiating both sides yields the following bound on the likelihood ratio:

$$\exp(-K\epsilon) \leq \frac{p_\theta(\hat{x}_0 \mid \tilde{x}_t)}{p_\theta(x_0 \mid \tilde{x}_t)} \leq \exp(K\epsilon). \tag{29}$$

This result shows that the likelihood ratio between the reconstructed sequence and the original sequence is constrained by their semantic similarity in the embedding space. Specifically, as the semantic discrepancy $\epsilon$ approaches zero, the likelihood ratio converges to 1, meaning that the reconstruction likelihood approaches that of the original sequence. This validates the use of semantic similarity as a proxy for likelihood stability, even in the presence of token-level variations such as paraphrasing. $\square$

### A.4. Manifold Distance Upper Bound (Proposition 3.5)

*Proof.* In this section, we derive a geometric error bound by modeling the denoising process as an operator on a metric space. Using the Banach Fixed Point Theorem, we show that the distance from a generated sample to the true solution is upper-bounded by the observable reconstruction drift.

Let $(\mathcal{Z}, d)$ be a complete metric space, where $\mathcal{Z}$ represents the embedding space of sequences and $d(x, y) = \|x - y\|$ is the Euclidean distance induced by the embedding norm. We define the denoising operator $\mathcal{T}_\theta : \mathcal{Z} \to \mathcal{Z}$ as the expected reconstruction:

$$\mathcal{T}_\theta(z) := \mathbb{E}_{\tilde{z} \sim q(\cdot|z)} \left[ \mathbb{E}_{z' \sim p_\theta(\cdot|\tilde{z})}[z'] \right]. \tag{30}$$

Let $\mathcal{M} = \{z \in \mathcal{Z} \mid \mathcal{T}_\theta(z) = z\}$ denote the set of valid fixed points. We assume that correct reasoning chains and their corresponding answers lie on this manifold $\mathcal{M}$. Let $z_0 \in \mathcal{Z}$ be an arbitrary generated sequence (candidate solution), and let $z^* \in \mathcal{M}$ be the closest valid solution to $z_0$, representing the ground truth.

We assume that the denoising operator $\mathcal{T}_\theta$ acts as a local contraction mapping around the manifold $\mathcal{M}$. That is, there exists a constant $0 \leq \kappa < 1$ such that for any $u, v \in \mathcal{Z}$ in the local neighborhood of $\mathcal{M}$:

$$\|\mathcal{T}_\theta(u) - \mathcal{T}_\theta(v)\| \leq \kappa\|u - v\|. \tag{31}$$

Our goal is to bound the true error $\|z_0 - z^*\|$ using only the observable reconstruction drift $\|z_0 - \mathcal{T}_\theta(z_0)\|$. By the triangle inequality, we have:

$$\|z_0 - z^*\| \leq \|z_0 - \mathcal{T}_\theta(z_0)\| + \|\mathcal{T}_\theta(z_0) - z^*\|. \tag{32}$$

Since $z^* \in \mathcal{M}$, it is a fixed point of the operator, so $\mathcal{T}_\theta(z^*) = z^*$. Thus, the second term becomes:

$$\|\mathcal{T}_\theta(z_0) - z^*\| = \|\mathcal{T}_\theta(z_0) - \mathcal{T}_\theta(z^*)\|. \tag{33}$$

Using the contraction mapping property:

$$\|\mathcal{T}_\theta(z_0) - \mathcal{T}_\theta(z^*)\| \leq \kappa\|z_0 - z^*\|. \tag{34}$$

Substituting this into the earlier inequality:

$$\|z_0 - z^*\| \leq \|z_0 - \mathcal{T}_\theta(z_0)\| + \kappa\|z_0 - z^*\|. \tag{35}$$

Rearranging to isolate $\|z_0 - z^*\|$:

$$(1 - \kappa)\|z_0 - z^*\| \leq \|z_0 - \mathcal{T}_\theta(z_0)\|. \tag{36}$$

Since $0 \leq \kappa < 1$, we can divide both sides by $(1 - \kappa)$ to obtain:

$$\|z_0 - z^*\| \leq \frac{1}{1 - \kappa}\|z_0 - \mathcal{T}_\theta(z_0)\|. \tag{37}$$

The term $\|z_0 - \mathcal{T}_\theta(z_0)\|$ represents the magnitude of the update vector proposed by the diffusion model during a single denoising step. This quantity is empirically estimated by the BMC score, which captures the negative semantic similarity. The above derivation shows that the distance to the true solution $z^*$ is upper-bounded by the reconstruction drift, scaled by $(1 - \kappa)^{-1}$. Thus, minimizing the reconstruction drift effectively minimizes the true error, providing a theoretical guarantee for the verification metric proposed by BMC. $\qquad\square$

### A.5. K-step Reconstruction

In practice, we perform reconstruction via a $K$-step estimation using the $x_0$-parameterized model $f_\theta$. We formally define the induced reconstruction distribution $P_\theta(x_0|\tilde{x}_t)$ via a reverse Markov chain parameterized by the denoiser $f_\theta$. Following the standard $x_0$-parameterization in discrete diffusion models (Austin et al., 2021), the single-step reverse transition probability is defined by marginalizing over the model's prediction of $x_0$:

$$p_\theta(x_{\tau_{i-1}}|x_{\tau_i}) = \sum_{\hat{x}_0 \in \mathcal{V}} q(x_{\tau_{i-1}}|x_{\tau_i}, \hat{x}_0)p_\theta(\hat{x}_0|x_{\tau_i}), \tag{38}$$

where $p_\theta(\hat{x}_0|x_{\tau_i})$ is the categorical distribution predicted by the neural network $f_\theta$ at step $\tau_i$, and $q(x_{\tau_{i-1}}|x_{\tau_i}, \hat{x}_0)$ is the tractable posterior of the forward process.

To generalize this to a $K$-step generative process, let $\{\tau_0, \tau_1, \ldots, \tau_K\}$ be a strictly increasing subsequence of timesteps such that $\tau_0 = 0$ and $\tau_K = t$. The induced reconstruction distribution marginalizes over all intermediate states along this trajectory:

$$P_\theta(x_0|\tilde{x}_t) := \sum_{x_{\tau_{K-1}},\ldots,x_{\tau_1}} \left[\prod_{i=1}^{K} p_\theta(x_{\tau_{i-1}}|x_{\tau_i})\right], \tag{39}$$

with boundary conditions $x_{\tau_K} = \tilde{x}_t$ and $x_{\tau_0} = x_0$.

**Remark:** In the special case where $K = 1$, the summation vanishes, and the distribution reduces to the direct single-step prediction, $P_\theta(x_0|\tilde{x}_t) = p_\theta(x_0|\tilde{x}_t)$. However, single-step prediction often results in high-bias approximations (e.g., averaging over modes) due to the multimodality of the posterior $q(x_0|\tilde{x}_t)$ in high-noise regimes. Crucially, for $K > 1$, the formulation marginalizes over intermediate trajectories, which introduces an iterative refinement mechanism. By decomposing the complex global mapping into a sequence of simpler local transitions, the multi-step process can progressively resolve ambiguity and correct quantization errors, leading to higher-fidelity reconstructions.

## B. Analysis of BMC Consistency Metrics

This appendix examines the six consistency metrics that constitute the BMC score. We present the rationale behind each metric, discuss their complementary functions, and explain the necessity of a multi-dimensional evaluation approach for reasoning verification.

### B.1. Multi-Dimensional Evaluation Framework

BMC employs six distinct metrics rather than a single scalar measurement. This design addresses the challenge of geometric reasoning verification, where the objective is to distinguish between stable manifold representations and superficial reconstruction patterns that may result from memorization or spurious correlations.

### B.1.1. LIMITATIONS OF SINGLE-METRIC APPROACHES

Single metrics prove inadequate for capturing three types of reconstruction errors.

Failure Mode 1: Fluent but Incorrect Reconstructions. A diffusion model trained on mathematical text may generate well-formatted reasoning chains with plausible intermediate steps while introducing logical errors. For example, a model might reconstruct "Janet sells 9 eggs at \$2 each, earning \$20" with high token-level accuracy (above 0.9) despite an arithmetic error. Token accuracy alone would indicate stability, though the final answer is incorrect.

Failure Mode 2: Semantic Divergence with Lexical Similarity. A reconstruction may preserve most tokens while altering critical words: "Janet sells 9 eggs..." becomes "Janet buys 9 eggs...". Token accuracy remains high (approximately 0.9), yet the semantic content has inverted. Lexical metrics alone cannot detect this divergence.

Failure Mode 3: Valid Paraphrasing Interpreted as Error. Mathematical reasoning permits multiple valid formulations. "Calculate $7 \times 3$" and "Compute $7 \times 3$" are semantically equivalent, yet token-level metrics penalize lexical variation. Exact matching criteria would incorrectly classify stable paraphrased reconstructions as errors.

These failure modes indicate that metrics must be sensitive to logical errors while remaining robust to valid linguistic variation. Single metrics cannot achieve this balance across diverse scenarios.

### B.1.2. HIERARCHICAL CONSISTENCY EVALUATION

BMC addresses this limitation through hierarchical evaluation across multiple granularities:

- **Token Level (Lexical):** Measures exact reconstruction fidelity via Token Accuracy ($s_{\text{tok}}$) and Character Similarity ($s_{\text{char}}$).
- **Embedding Level (Semantic):** Captures meaning preservation through Semantic Similarity ($s_{\text{sem}}$).
- **Structure Level (Logical Nodes):** Tracks critical reasoning components via Number Retention ($s_{\text{num}}$).
- **Outcome Level (Terminal Answer):** Verifies solution correctness through Final Answer Match ($s_{\text{ans}}$).
- **Density Level (Confidence):** Probes manifold position via Intrinsic Confidence ($s_{\text{conf}}$).

Aggregating signals across these levels produces a consistency measure that tolerates valid paraphrases without accepting fluent hallucinations.

### B.2. Metric Definitions

### B.2.1. TOKEN ACCURACY

Token Accuracy quantifies exact reconstruction by measuring whether the model assigns dominant probability mass to the original tokens during reconstruction. This metric serves as the lexical baseline for consistency evaluation.

Under the manifold hypothesis (Proposition 3.5), a valid solution $x_0$ should act as a strong attractor. Token Accuracy measures the local strength of this attractor: if $\mathcal{T}_\theta$ is contractive, it should recover the precise original configuration from perturbed states.

Token Accuracy is particularly relevant for mathematical reasoning due to the non-interchangeability of numerical symbols. Unlike natural language, where synonyms may be substituted without changing meaning, numerical values admit no such flexibility.

This metric operationalizes probabilistic concentration. High token accuracy indicates that the original sequence lies in a sharp probability peak of the learned distribution. It provides a necessary filter against subtle errors where a single digit change (e.g., $100 \to 10$) invalidates the entire reasoning chain despite high semantic similarity.

The primary limitation of Token Accuracy is its lexical rigidity. The metric penalizes valid paraphrasing, resulting in false negatives. Consider semantically equivalent pairs where Token Accuracy assigns low scores:

- *Synonymy:* "**Compute** $7 \times 3$" vs. "**Calculate** $7 \times 3$" (Score: $\approx 0.0$)
- *Formatting:* "Answer: **18**" vs. "Answer: **18.0**" (Score: $\approx 0.5$)
- *Ordering:* "**First**, subtract 5" vs. "**Initially**, subtract 5" (Score: $\approx 0.0$)

In these cases, the strict exact-match requirement misclassifies linguistic flexibility as geometric instability.

Table 4 demonstrates this limitation. Token Accuracy yields an AUROC of approximately 0.64 on GSM8K, compared to BMC's 0.89. The 25-point gap reflects the metric's inability to accommodate natural language variation, motivating the use of complementary semantic and structural metrics.

### B.2.2. SEMANTIC SIMILARITY

Semantic Similarity evaluates whether the meaning of the reconstruction matches the original, independent of specific lexical choices. This metric encodes both sequences into a shared embedding space using a pre-trained sentence transformer and computes the cosine similarity of their representations.

The metric operationalizes semantic equivalence, determining whether two sequences express the same underlying logic despite surface-level variations in phrasing.

The theoretical basis derives from Proposition 3.4 (Semantic Continuity of Likelihood), which establishes a connection between geometric distance in the embedding space and probabilistic stability. The proposition asserts that if the log-likelihood function is locally Lipschitz continuous, then for any reconstruction $\hat{x}_0$ semantically close to the original $x_0$ (i.e., $d_\Phi(x_0, \hat{x}_0) \leq \epsilon$), the likelihood ratio remains bounded within an exponential envelope determined by the Lipschitz constant and semantic distance.

This bound provides rigorous justification for semantic relaxation. It ensures that a reconstruction with high semantic similarity (small $\epsilon$) retains comparable probability mass to the original, rather than representing an arbitrary hallucination. Unlike strict token matching, which requires the model to collapse onto a single mode, this continuity allows probability mass to be distributed over a local neighborhood of semantically equivalent phrasings.

Geometrically, Semantic Similarity recognizes that the validity manifold $\mathcal{M}$ comprises multiple linguistic formulations that map to the same logical reasoning, rather than a single trajectory.

Semantic Similarity addresses cases where Token Accuracy is overly restrictive. Consider a paraphrasing example from GSM8K:

*Original:* "Janet sells the remaining 9 eggs at the farmer's market for $2 each, earning $18 daily."
*Reconstructed:* "She vends the leftover 9 eggs at the market for $2 apiece, making $18 per day."

Token Accuracy assigns approximately 0.3 because only articles and numbers align. However, since the semantic content is preserved, Semantic Similarity correctly assigns a score of approximately 0.93, preventing the false negative that would result from relying solely on lexical matching.

Despite its robustness to phrasing variations, Semantic Similarity may exhibit semantic collapse. Since embedding models are trained on general linguistic corpora, they may assign high similarity scores to fluent but factually incorrect text, particularly when numerical values change. For example:

*Original:* "The answer is **18** dollars."
*Reconstructed:* "The answer is **81** dollars."

A generic embedding model might assign these sentences a high similarity score (approximately 0.85) due to identical syntactic structures and semantic categories. However, the numerical content has diverged.

BMC mitigates this vulnerability through metric cross-validation. When Semantic Similarity fails to detect numerical changes, Number Retention flags the mismatch, and Final Answer Match provides a definitive signal. This multi-layered approach prevents semantic collapse from compromising overall verification.

### B.2.3. NUMBER RETENTION

Number Retention quantifies the preservation of numerical values within the reasoning chain. Unlike natural language tokens which may admit synonymous substitution, numerical literals in mathematical reasoning serve as rigid constraints that define the solution space.

This metric operationalizes numerical consistency: a valid reconstruction must preserve the exact set of numerical values present in the original trajectory to maintain logical integrity.

Numbers function as integrity checkpoints along the reasoning manifold. Consider a typical GSM8K problem:

*Context:* "Janet's ducks lay 16 eggs... She eats 3... bakes 4..."
*Reasoning:* Requires tracking the set $\{16, 3, 4, 9\}$.

If a reconstruction alters any of these values (e.g., "She eats 5 for breakfast"), the entire reasoning chain is invalidated, regardless of linguistic fluency. Unlike semantic concepts (where "compute" and "calculate" are interchangeable), numerical values possess exact semantics with no valid paraphrasing that preserves arithmetic correctness. Number Retention detects these violations that Semantic Similarity might overlook.

Number Retention can detect early drift in reasoning chains. Errors often manifest as numerical inconsistencies in intermediate steps before the final answer. For instance:

*Original:* "Step 1: $16 - 3 = 13$. Step 2: $13 - 4 = 9$. Answer: 9"
*Reconstructed:* "Step 1: $16 - 3 = 13$. Step 2: $13 - \mathbf{5} = \mathbf{8}$. Answer: **8**"

Number Retention flags the spurious introduction of "5" and the divergence of the intermediate result "8", indicating that the trajectory has drifted from the validity manifold during the reasoning process. This provides a finer-grained stability signal than final answer checking alone.

A naive recall-based metric would fail to penalize numerical hallucination. Consider a case where the model introduces irrelevant calculations:

*Original:* "$16 - 7 = 9$. Answer: 9"
*Reconstructed:* "$16 - 7 = 9$. Also, $9 \times 100 = \mathbf{900}$. But the answer is 9."

Although all original numbers are retained (Recall = 1.0), the reconstruction demonstrates instability by introducing irrelevant computational paths. BMC can optionally incorporate a precision penalty where the score is reduced for each surplus number introduced. When enabled, this penalty discourages hallucinations while permitting legitimate intermediate derivations.

### B.2.4. CHARACTER SIMILARITY

Character Similarity operates at the sub-lexical level by computing the normalized Levenshtein edit distance between the original and reconstructed sequences. This metric quantifies morphological consistency, measuring the structural preservation of text independent of the discrete tokenization boundaries imposed by the model's vocabulary.

Character Similarity occupies a position between the binary structure of Token Accuracy and the continuous semantic space of Semantic Similarity. Its primary function is to provide robustness against tokenization artifacts.

Standard BPE or WordPiece tokenizers often split related words unpredictably. Character Similarity detects shared morphological structure that Token Accuracy misses. For example, "calculate" (1 token) versus "calculation" (1 different token) yields Token Accuracy of 0.0 but Character Similarity of approximately 0.83 due to high structural overlap.

Mathematical notation is prone to inconsistent tokenization. The representation "$18" might be tokenized as ['$', '18'], while "18 dollars" becomes ['18', 'dollars']. While these sets share little lexical overlap, their character-level structures remain highly similar. This metric ensures that valid formatting variations are not penalized.

Minor typographical errors (e.g., "occured" versus "occurred") can result in disjoint token IDs. Character Similarity provides graceful degradation, preventing single errors from eliminating the consistency signal.

Despite its utility, Character Similarity exhibits surface-level bias: it can assign high scores to semantically divergent text that shares character sequences. Consider the following case:

*Original:* "The answer is **18** dollars."
*Reconstructed:* "The answer is **81** dollars."

Because only two characters are transposed ("18" versus "81"), Character Similarity remains high (approximately 0.93), despite the logical error.

Due to this risk, Character Similarity serves as an auxiliary metric with specific functions: (1) tiebreaking when other metrics are ambiguous, distinguishing structural variants from complete hallucinations; (2) smoothing by providing non-zero

gradients when tokenization artifacts drive Token Accuracy to zero; (3) complementarity by revealing morphological consistency that token boundaries obscure. Ablation on GSM8K shows measurable but modest impact, confirming its role in refining boundary cases rather than primary discrimination.

### B.2.5. FINAL ANSWER MATCH

Final Answer Match evaluates terminal convergence of the generation process by determining whether the reconstructed trajectory reaches the same solution as the original. Unlike continuous metrics that measure path similarity, this metric implements a binary consistency constraint: $s_{\mathrm{ans}} \in \{0, 1\}$. It provides explicit grounding that links geometric stability to task-specific correctness.

From the geometric perspective in Proposition 3.5, the final answer represents the projection of the reasoning trajectory onto the discrete solution manifold. The manifold error bound establishes that distance to the true solution is inversely proportional to model consistency, scaled by reconstruction drift. If extracted answers differ (i.e., $E(z_0) \neq E(\hat{z}_0)$), the reconstruction drift $\|z_0 - \mathcal{T}_\theta(z_0)\|$ exceeds a local perturbation and crosses the decision boundary between distinct attractor basins.

This phenomenon, termed basin hopping, constitutes the geometric signature of instability. Even if a trajectory maintains high lexical overlap, divergence in the terminal state indicates that the denoising operator fails to contract toward a single fixed point. Final Answer Match thus provides a necessary condition for stability: a solution cannot be considered stable if its terminal conclusion varies.

The validity of a binary metric depends on its extraction logic. Naive matching (e.g., finding the last number) is prone to noise. To ensure that $s_{\mathrm{ans}}$ reflects logical convergence rather than formatting artifacts, BMC employs a hierarchical extraction strategy based on information rigidity.

**For Multiple-Choice Tasks (GPQA, ARC).** We prioritize explicit markup over free text to handle verbose outputs:

1. Tier 1 (Explicit Markup): Tagged content (e.g., \boxed{A}, <answer>A</answer>).
2. Tier 2 (Declarative Statements): Templated phrases (e.g., "The answer is A").
3. Tier 3 (Heuristic Fallback): Isolated tokens at the end of the text (used only when high-priority signals are absent).

**For Numerical Tasks (GSM8K, MATH).** We filter intermediate calculations to isolate the terminal value:

1. Tier 1 (Structural Tags): <answer>42</answer> or dataset-specific delimiters (e.g., #### 42).
2. Tier 2 (Semantic Assertions): Explicit concluding statements (e.g., "The final answer is 42").
3. Tier 3 (Numeric Suffix): The last valid numerical literal, penalized if the chain contains multiple unformatted numbers.

This hierarchy ensures comparison of intended outputs, making the metric robust to formatting variations between forward and backward passes.

The effectiveness of Final Answer Match is demonstrated in ablation studies (Table 4). Using $s_{\mathrm{ans}}$ alone achieves an AUROC of 0.819 on GSM8K, outperforming any other individual metric (including Token Accuracy at 0.640). Removing this metric from the composite score produces the largest performance degradation (13.6 percentage points), confirming that terminal convergence captures a substantial portion of the stability signal.

While it is the strongest single predictor, Final Answer Match remains insufficient independently because it cannot detect cases where the correct answer follows from incorrect reasoning. It therefore functions as the dominant component within the multi-dimensional BMC framework, supported by semantic and structural verification.

### B.2.6. INTRINSIC CONFIDENCE

Intrinsic Confidence is the framework's only generative metric, derived from the model's internal predictive distribution without reference to external ground truth. It aggregates decisional certainty across the iterative denoising trajectory:

$$s_{\mathrm{conf}} = \frac{1}{K} \sum_{k=1}^{K} \left( \frac{1}{|M_k|} \sum_{i \in M_k} \max_{v \in \mathcal{V}} p_\theta(v | \tilde{x}_t^{(k)}) \right). \tag{40}$$

The inner term computes average peak probability over the masked set $M_k$ at step $k$, while the outer summation averages across reverse diffusion steps. This two-level aggregation captures the evolving uncertainty of the generation process,

measuring the model's decisiveness as it progressively resolves the output.

Theoretically, Intrinsic Confidence serves as a computationally efficient proxy for manifold density. Grounded in the probabilistic framework of Proposition 3.2, maximizing the BMC objective approximates the Evidence Lower Bound (ELBO), where the term $\mathbb{E}[\log p_\theta(x_0|x_t)]$ assigns higher scores to high-probability configurations.

High confidence ($s_{\text{conf}} \to 1$) implies the trajectory resides in a high-likelihood region of the learned distribution. Conversely, low confidence signals high entropy, indicating the trajectory has drifted to a region where the model lacks clear reconstruction direction.

Despite its theoretical connection to density, this metric cannot serve as a primary verifier due to calibration limitations. Diffusion models trained with maximum likelihood often exhibit template memorization, assigning high probability to frequent syntactic patterns regardless of logical validity.

**Failure Case: Confident Hallucination.**

*Original:* "$16 - 7 = 9$. Answer: 9"
*Reconstructed:* "$16 - 7 = 10$. Answer: 10"
*Metric:* $s_{\text{conf}} = 0.84$ (High Confidence, Wrong Answer)

The model confidently reconstructs the error because it recognizes the syntactic template "$X - Y = Z$", treating numbers as interchangeable slots. High confidence thus reflects consistency with training statistics rather than logical correctness.

Given these calibration issues, Intrinsic Confidence functions as a secondary validator within the BMC framework. When structural metrics indicate correctness ($s_{\text{ans}} = 1.0$), high confidence reinforces the stability assessment, confirming the solution is probabilistically robust. Extremely low confidence ($s_{\text{conf}} < 0.5$) serves as a signal of manifold collapse, filtering cases where the model fails to find coherent paths.

### B.3. Complementarity Analysis

The six-metric design provides robust verification through hierarchical validation. Ablation studies (Table 4) confirm that removing any metric degrades performance. Final Answer Match is the strongest single component (AUROC drops from 0.889 to 0.753 when removed), Number Retention is essential for mathematical tasks (drop to 0.821), Semantic Similarity prevents false negatives from lexical rigidity (drop to 0.850), and auxiliary metrics (Character Similarity, Intrinsic Confidence) contribute measurably (approximately 0.880 when removed). No subset of five metrics replicates the full framework's discriminative power (AUROC 0.82 to 0.90 across benchmarks).

This complementarity addresses three fundamental failure modes through hierarchical validation. Fluent hallucinations are detected by Number Retention and Final Answer Match. Semantic drift is identified by Semantic Similarity despite high Token Accuracy. Valid paraphrasing is preserved without penalty. Primary metrics (Final Answer, Number Retention) provide strong signals, secondary metrics (Semantic and Character Similarity) handle edge cases, and auxiliary signals (Intrinsic Confidence) refine ranking, ensuring that no single calibration failure compromises overall verification.

## C. Implementation and Complexity

### C.1. Baseline Implementation

This section provides implementation details for the error diagnosis baselines evaluated in Table 1.

**Model Confidence for Diffusion Language Models.** Unlike AR models that produce sequential likelihoods $p(x) = \prod_{i=1}^{L} p(x_i|x_{<i})$, diffusion language models generate text through iterative denoising with dynamic unmasking. We compute Model Confidence by averaging the predicted probabilities throughout the denoising process.

For each token position $i$ in the generated sequence, we track all diffusion steps where position $i$ was unmasked (i.e., transitioned from [MASK] to a concrete token). Let $U_i$ denote this set of unmasking steps. The position-level confidence is computed as:

$$\text{Confidence}_i = \frac{1}{|U_i|} \sum_{t \in U_i} p_\theta(x_i|x_t^{(\backslash i)}, t), \tag{41}$$

where $x_i$ is the final token at position $i$, and $x_t^{(\backslash i)}$ denotes the partial sequence at step $t$ excluding position $i$. Note that $|U_i|$ may exceed 1 due to remasking strategies employed by some diffusion models.

The final Model Confidence score aggregates across all positions:

$$\text{Model Confidence} = \frac{1}{L} \sum_{i=1}^{L} \text{Confidence}_i. \tag{42}$$

This formulation captures the model's average decisiveness in token predictions throughout denoising, analogous to sequence likelihood in AR models but adapted to the bidirectional generation dynamics of diffusion (Song et al., 2020; Gao et al., 2025).

**Self-Consistency.** Following Wang et al. (Wang et al., 2022), we generate $N = 10$ independent samples per query using the model's standard generation procedure. Let $\mathcal{S} = \{x^{(1)}, \ldots, x^{(N)}\}$ be the sample set. We extract the final answer from each sample using task-specific extractors $\mathcal{E}(\cdot)$ and identify the majority answer:

$$a^* = \arg\max_a |\{x \in \mathcal{S} : \mathcal{E}(x) = a\}|. \tag{43}$$

To derive a continuous confidence score suitable for AUROC/AUPR computation, we use the vote ratio:

$$\text{SC}(q) = \frac{|\{x \in \mathcal{S} : \mathcal{E}(x) = a^*\}|}{N} \in [0, 1], \tag{44}$$

where a score of 1.0 indicates unanimous agreement across all samples, while lower scores indicate disagreement. For example, with $N = 10$, a score of 0.5 indicates a tie between the top two answers.

**Self-Evaluation.** Self-Evaluation leverages the model's metacognitive ability to assess its own correctness (Madaan et al., 2023). After generating a solution $x_0$ for query $q$, we construct an evaluation prompt:

> *Question:* [original query $q$]
>
> *Your Answer:* [generated solution $x_0$]
>
> *Is this answer correct? Respond with a confidence score between 0.0 (definitely wrong) and 1.0 (definitely correct). Only output the numerical score, nothing else.*

We then feed this prompt to the model and extract the generated response. The model may produce various formats (e.g., "0.85", "8.5", or natural language containing a score). We use regular expression matching to extract the first numerical value:

$$s_{\text{raw}} = \text{extract\_number}(\text{model\_response}). \tag{45}$$

Since some models output scores on a 0–10 scale, we normalize to $[0, 1]$:

$$\text{Self-Eval}(q, x_0) = \begin{cases} s_{\text{raw}}/10 & \text{if } s_{\text{raw}} > 1.0 \\ s_{\text{raw}} & \text{otherwise} \end{cases}. \tag{46}$$

The final score is clipped to $[0, 1]$ to ensure validity. This approach enables the model to provide graded confidence assessments rather than binary judgments, facilitating continuous evaluation metrics.

## C.2. Complexity Analysis

In this section, we substantiate the efficiency claims by analyzing the computational cost in terms of diffusion denoising steps. Let $T$ denote the number of steps for full generation and $K$ for backward reconstruction. A key property of our method is that verification is truncated, such that $K \ll T$.

**1. Analysis for Error Diagnosis.** In the error diagnosis setting, we compare the cost of establishing a validity signal.

- **Self-Consistency (SC):** Standard SC relies on an ensemble of $N_{\text{SC}}$ samples (typically $N_{\text{SC}} \geq 10$). The total cost is:

$$\mathcal{C}_{\text{SC}} = N_{\text{SC}} \times T. \tag{47}$$

- **BMC Verification:** Our method generates a single sample and verifies it using an ensemble of $N_{\text{BMC}}$ reconstruction steps. The total cost is:

$$\mathcal{C}_{\text{BMC}} = 1 \times T + N_{\text{BMC}} \times K. \tag{48}$$

**Conclusion:** In our experiments, we use a small ensemble $N_{\text{BMC}} = 4$. Since $K \ll T$ (specifically $K \approx 0.015T$), the verification term $N_{\text{BMC}}K$ is negligible. Consequently, $\mathcal{C}_{\text{BMC}} \approx T \ll N_{\text{SC}}T = \mathcal{C}_{\text{SC}}$. This demonstrates that BMC provides discriminative error diagnosis at a fraction of the cost of standard Self-Consistency.

**2. Analysis for Adaptive Self-Correction.** In the inference setting, we compare the total compute required to find a solution.

- **Fixed-Budget Baseline (SC):** SC uses a fixed sample budget $N_{\text{SC}}$.

$$\mathcal{C}_{\text{SC}} = N_{\text{SC}} \times T. \tag{49}$$

- **MGRS Adaptive Inference:** Our method employs an iterative process. The total cost depends on the average number of samples $N_{\text{avg}}$:

$$\mathcal{C}_{\text{BMC}} = N_{\text{avg}} \times (T + N_{\text{BMC}} \times K). \tag{50}$$

**Conclusion:** Again, due to $K \ll T$, the marginal overhead of verification is minimal, implying $(T + N_{\text{BMC}}K) \approx T$. The efficiency comparison simplifies to comparing the sample counts $N_{\text{avg}}$ versus $N_{\text{SC}}$. Since BMC enables "early stopping" on simpler problems, $N_{\text{avg}}$ is typically much lower than the fixed budget required for robust SC (i.e., $N_{\text{avg}} < N_{\text{SC}}$). Therefore, BMC achieves superior performance with significantly lower total computational cost.

### C.3. Hyperparameter Settings

This section describes the hyperparameter selection for two downstream applications of BMC: Manifold-Guided Rejection Sampling (MGRS, §4.2) and RL Alignment (§4.3).

**MGRS Threshold Selection ($\tau = 0.75$).** The acceptance threshold controls the trade-off between solution quality and sampling efficiency. We select $\tau = 0.75$ by tuning on a held-out validation set from GSM8K to maximize the F1 score between precision (the fraction of accepted solutions that are correct) and recall (the fraction of correct solutions that are accepted). Empirically, correct solutions exhibit mean BMC score $\approx 0.85$ while incorrect ones average $\approx 0.60$, providing a clear discriminative signal. The threshold $\tau = 0.75$ achieves precision $> 85\%$ and recall $> 80\%$, balancing early-stopping efficiency with solution quality. This threshold generalizes across reasoning benchmarks without dataset-specific tuning.

**RL Reward Design ($r_{\text{base}} = 1.5$).** We set the baseline reward $r_{\text{base}} = 1.5$ to maintain compatibility with the original outcome-supervised framework, which assigns reward 2.0 to correct solutions. With $\alpha_t \in [0.5, 1.0]$ and BMC scores typically $\in [0, 1]$, the BMC-augmented reward ranges from 1.5 to 2.5, yielding an expected reward of $\approx 2.0$ when $S_{\text{BMC}} \approx 0.5$. This design preserves training stability (similar reward scale) while providing fine-grained differentiation between high and low geometric stability within correct solutions.

**Annealing Schedule ($\alpha_{\min} = 0.5$, $\alpha_{\max} = 1.0$).** The linear schedule $\alpha_t = 0.5 + 0.5 \cdot \frac{t}{T}$ gradually shifts emphasis from answer correctness (early training, low $\alpha_t$) to geometric stability (late training, high $\alpha_t$), preventing premature convergence to high-stability but incorrect solutions. The training budget $T$ is set to 6,000 iterations for GSM8K and 4,000 iterations for MATH, ARC-Challenge, and GPQA, balancing convergence quality with computational efficiency.

## D. Additional Experimental Results

### D.1. Extended Baseline

Table 5 extends Table 2 to include $N = 10$ baselines for both LLaDA-8B-Instruct and Dream-v0-Instruct-7B. While Self-Consistency with $N = 10$ samples achieves slightly higher accuracy on some tasks, MGRS uses significantly fewer

*Table 5.* **Full Comparison of Adaptive Self-Correction Methods.** Extended version of Table 2 including $N = 10$ baselines. MGRS adaptively stops when BMC score exceeds $\tau = 0.75$ (up to $N_{\max} = 10$).

| Model | Method | GSM8K | | MATH | | ARC | | GPQA | |
|---|---|---|---|---|---|---|---|---|---|
| | | Acc↑ | Eff↑ | Acc↑ | Eff↑ | Acc↑ | Eff↑ | Acc↑ | Eff↑ |
| LLaDA | Single-pass | 70.5 | - | 24.4 | - | 83.2 | - | 24.8 | - |
| | Self-Consistency ($N$=3) | 74.3 | 1.86 | 24.2 | -0.10 | 86.1 | 1.45 | 25.0 | 0.11 |
| | Self-Consistency ($N$=10) | 79.4 | 0.98 | 28.2 | 0.42 | 88.7 | 0.61 | 27.9 | 0.35 |
| | Best-of-N Confidence ($N$=3) | 70.7 | 0.08 | 23.8 | -0.30 | 83.3 | 0.04 | 27.7 | 1.45 |
| | Best-of-N Confidence ($N$=10) | 71.1 | 0.06 | 23.8 | -0.07 | 82.1 | -0.12 | 22.3 | -0.27 |
| | Best-of-N BMC ($N$=3) | 77.8 | 3.64 | 26.0 | 0.80 | 86.3 | 1.54 | 25.0 | 0.11 |
| | Best-of-N BMC ($N$=10) | 78.7 | 0.91 | **29.0** | 0.51 | 85.2 | 0.22 | 26.6 | 0.20 |
| | **MGRS** | **79.5** | **3.98** | 27.6 | **0.66** | 87.2 | **1.85** | 26.1 | 0.49 |
| Dream | Single-pass | 72.0 | - | 22.4 | - | 74.7 | - | 22.3 | - |
| | Self-Consistency ($N$=3) | 72.4 | 0.19 | 20.2 | -1.10 | 80.6 | 2.91 | 29.5 | 3.57 |
| | Self-Consistency ($N$=10) | 72.5 | 0.05 | 20.4 | -0.22 | 80.6 | 0.65 | 29.7 | 0.82 |
| | Best-of-N Confidence ($N$=3) | 72.5 | 0.23 | 20.2 | -1.10 | 80.4 | 2.82 | 29.5 | 3.57 |
| | Best-of-N Confidence ($N$=10) | 72.5 | 0.05 | 20.0 | -0.27 | 80.4 | 0.63 | 28.8 | 0.72 |
| | Best-of-N BMC ($N$=3) | 72.5 | 0.23 | 20.2 | -1.10 | 80.5 | 2.86 | 29.7 | **3.68** |
| | Best-of-N BMC ($N$=10) | 72.5 | 0.05 | 20.2 | -0.24 | 80.4 | 0.63 | **29.9** | 0.84 |
| | **MGRS** | **72.9** | **0.69** | **22.6** | **0.05** | **81.0** | **3.21** | 27.9 | 1.20 |

*Table 6.* **Average Sample Count for MGRS Method.** Average number of samples used by MGRS across different datasets and models.

| Model | GSM8K | MATH | ARC | GPQA |
|---|---|---|---|---|
| LLaDA | 3.3 | 5.8 | 3.2 | 3.8 |
| Dream | 2.2 | 5.4 | 2.9 | 5.7 |

samples on average (Table 6), resulting in better sample efficiency. Notably, Best-of-N Confidence with $N = 10$ often performs worse than $N = 3$, confirming that confidence-based selection does not benefit from additional samples.

Several key observations emerge from this extended comparison:

**Diminishing returns of additional samples.** On LLaDA, increasing Self-Consistency from $N = 3$ to $N = 10$ improves GSM8K accuracy from 0.743 to 0.794 but reduces sample efficiency from 1.86 to 0.98. MGRS achieves 0.795 using only 3.3 samples on average (Table 6). On Dream, the efficiency drop is even more severe (from 0.19 to 0.05 on GSM8K), while MGRS maintains better efficiency (0.69) despite using only 2.2 samples on average.

**Confidence-based selection fails with more samples.** Best-of-N Confidence with $N = 10$ consistently underperforms $N = 3$ across both models. On LLaDA GPQA, accuracy drops from 0.277 to 0.223 (5.4% decrease). On Dream GPQA, efficiency drops from 3.57 to 0.72. This confirms that token confidence is unreliable and does not benefit from additional sampling.

**MGRS achieves competitive accuracy with fewer samples.** While some $N = 10$ baselines achieve slightly higher accuracy on specific tasks (e.g., LLaDA MATH: SC 0.282 vs MGRS 0.276; Dream GPQA: Best-of-N BMC 0.299 vs MGRS 0.279), MGRS uses 40-60% fewer samples on average, resulting in superior overall efficiency. As shown in Table 6, MGRS typically requires 2-6 samples across different datasets, with higher sample counts only on challenging tasks like MATH (5.82 for LLaDA, 5.41 for Dream). The adaptive stopping mechanism successfully balances accuracy and computational cost, automatically allocating more samples to harder problems while stopping early on easier ones.

**Task-dependent sampling behavior.** Table 6 reveals interesting patterns in MGRS's sampling behavior. MATH consistently requires the most samples (5.8 for LLaDA, 5.4 for Dream), reflecting its difficulty. In contrast, GSM8K requires fewer samples (3.3 for LLaDA, 2.2 for Dream), indicating that BMC can identify correct solutions earlier for these problems. This adaptive behavior is a key advantage over fixed-N methods, which cannot adjust to problem difficulty.

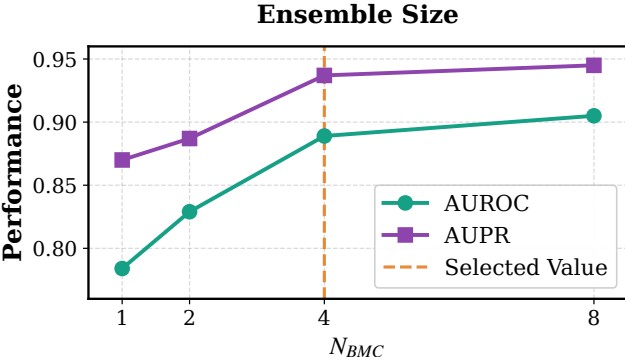

*Figure 4.* **Effect of Ensemble Size on Error Detection Performance.** Both AUROC and AUPR improve monotonically with ensemble size $N_{\text{BMC}}$, exhibiting standard variance reduction behavior. Performance saturates beyond $N_{\text{BMC}} = 4$, with marginal gains at higher computational cost. The dashed orange line indicates our selected default value.

### D.2. Ablation Studies

We conduct systematic ablation studies on three key hyperparameters: backward diffusion steps $K$, masking ratio $\gamma$, and ensemble size $N_{\text{BMC}}$. Table 7 presents complete results including computational costs measured relative to the baseline configuration ($K = 4$, $N_{\text{BMC}} = 1$). These results complement the visualizations in Figure 3.

*Table 7.* Complete Hyperparameter Ablation on GSM8K. Default configuration: $K = 16$, $\gamma = 0.9$, $N_{\text{BMC}} = 4$.

| Configuration | AUROC | AUPR | Cost |
|---|---|---|---|
| *Backward Diffusion Steps ($K$)* | | | |
| $K = 4$ steps | 0.840 | 0.913 | 1× |
| $K = 16$ steps | **0.873** | **0.923** | 4× |
| $K = 64$ steps | 0.875 | 0.929 | 16× |
| $K = 128$ steps | 0.879 | 0.928 | 32× |
| *Masking Ratio ($\gamma$)* | | | |
| $\gamma = 0.0$ (no mask) | 0.598 | 0.753 | - |
| $\gamma = 0.3$ (30%) | 0.672 | 0.794 | - |
| $\gamma = 0.5$ (50%) | 0.712 | 0.816 | - |
| $\gamma = 0.7$ (70%) | 0.794 | 0.873 | - |
| $\gamma = 0.9$ (90%) | **0.889** | **0.937** | - |
| $\gamma = 1.0$ (full mask) | 0.712 | 0.861 | - |
| *Ensemble Size ($N_{BMC}$)* | | | |
| $N_{\text{BMC}} = 1$ sample | 0.784 | 0.870 | 1× |
| $N_{\text{BMC}} = 2$ samples | 0.829 | 0.887 | 2× |
| $N_{\text{BMC}} = 4$ samples | **0.889** | **0.937** | 4× |
| $N_{\text{BMC}} = 8$ samples | 0.905 | 0.945 | 8× |

**Backward Diffusion Steps ($K$).** Performance saturates at $K = 16$ (AUROC 0.873, cost 4×), validating our theoretical claim that truncated reconstruction suffices for error detection. Increasing to $K = 64$ yields only +0.002 AUROC improvement at 16× cost, while $K = 128$ offers minimal additional gains (+0.004 AUROC) at 32× cost. This confirms that the manifold geometry relevant to correctness assessment is captured within the first 16 backward steps, beyond which we only refine low-level token details that do not affect semantic consistency.

**Masking Ratio ($\gamma$).** The inverted-U pattern peaks at $\gamma = 0.9$ (AUROC 0.889), confirming the importance of balancing reconstruction difficulty and geometric anchoring. No masking ($\gamma = 0.0$) yields poor performance (AUROC 0.598) as the reconstruction task becomes trivial. Conversely, full masking ($\gamma = 1.0$) degrades performance to 0.712 AUROC by removing all geometric anchors, forcing the model to hallucinate content without grounding. The optimal $\gamma = 0.9$ creates sufficient reconstruction challenge while preserving 10% of tokens as geometric anchors to constrain the backward trajectory.

**Ensemble Size ($N_{\text{BMC}}$).** Figure 4 visualizes the effect of ensemble size on error detection performance. Both AUROC and AUPR improve monotonically with $N_{\text{BMC}}$, exhibiting standard variance reduction behavior. However, the improvement

shows diminishing returns: while moving from $N_{\text{BMC}} = 1$ to $N_{\text{BMC}} = 4$ yields substantial gains (+10.5 AUROC points, +6.7 AUPR points), further doubling to $N_{\text{BMC}} = 8$ provides only marginal improvement (+1.6 AUROC points, +0.8 AUPR points) at 2× computational cost.

This behavior aligns with standard Monte Carlo estimation theory, where estimation error decreases at rate $O(1/\sqrt{N_{\text{BMC}}})$. We select $N_{\text{BMC}} = 4$ as the default configuration to balance performance and efficiency: it achieves stable performance (AUROC 0.889, AUPR 0.937) while keeping computational overhead manageable for practical deployment. For applications where computational budget is less constrained, $N_{\text{BMC}} = 8$ can be used to achieve near-optimal performance (AUROC 0.905, AUPR 0.945). Conversely, for extremely fast inference, even $N_{\text{BMC}} = 1$ provides reasonable discriminative power (AUROC 0.784), though with higher variance in individual predictions.

### D.3. Cross-Domain Generalization to Code Generation

While mathematical reasoning relies on numerical constants, open-ended code generation requires evaluating structural logic. To demonstrate the generalizability of the BMC framework, we evaluate an alternative instantiation of $\mathcal{D}$ on programming tasks, using Abstract Syntax Tree (AST) Jaccard similarity and token substitution rates to capture syntactic stability without terminal answer tokens.

As evaluated on HumanEval and MBPP in Table 8, BMC consistently outperforms standard likelihood-based and self-consistency baselines. This confirms that the underlying perturb-and-reconstruct protocol operates independently of domain-specific text formatting.

*Table 8.* Discriminative Performance of BMC on Open-Ended Code Generation Tasks.

| Method | HumanEval | | MBPP | |
|---|---|---|---|---|
| | AUROC ↑ | AUPR ↑ | AUROC ↑ | AUPR ↑ |
| Model Confidence | 0.535 | 0.630 | 0.528 | 0.581 |
| Self-Consistency | 0.615 | 0.684 | 0.651 | 0.669 |
| Self-Evaluation | 0.683 | 0.727 | 0.639 | 0.646 |
| **BMC (Ours)** | **0.687** | **0.783** | **0.662** | **0.668** |

### D.4. Evaluation on Alternative Architectures and Scales

To verify the robust manifestation of trajectory stability across diverse model parameters, we report supplementary error diagnosis results on an MoE model (LLaDA-MoE-7B-A1B-Instruct) and a larger dense architecture (LLaDA2.1-mini). As summarized in Table 9, BMC maintains consistent discriminative capacities across variations in attention mechanisms and parameter scales.

*Table 9.* Error Diagnosis Performance (GSM8K) across different dLLM architectures and scales.

| Model | Architecture | AUROC ↑ | AUPR ↑ |
|---|---|---|---|
| LLaDA-8B-Instruct | Dense, 8B | 0.893 | 0.943 |
| Dream-v0-Instruct-7B | Dense, 7B | 0.898 | 0.787 |
| LLaDA-MoE-7B-A1B-Instruct | MoE, 7B (1B active) | 0.858 | 0.878 |
| LLaDA2.1-mini | Dense, 16B | 0.853 | 0.933 |

## E. Geometric and Qualitative Analysis

### E.1. Geometric Validation

#### E.1.1. MOTIVATION: LOCAL CONTRACTION

Proposition 3.5 assumes that the valid solution manifold $\mathcal{M}$ acts as a stable attractor where the denoising operator is locally contractive ($\kappa \approx 0$). To validate this assumption, we examine the relationship between manifold density (how

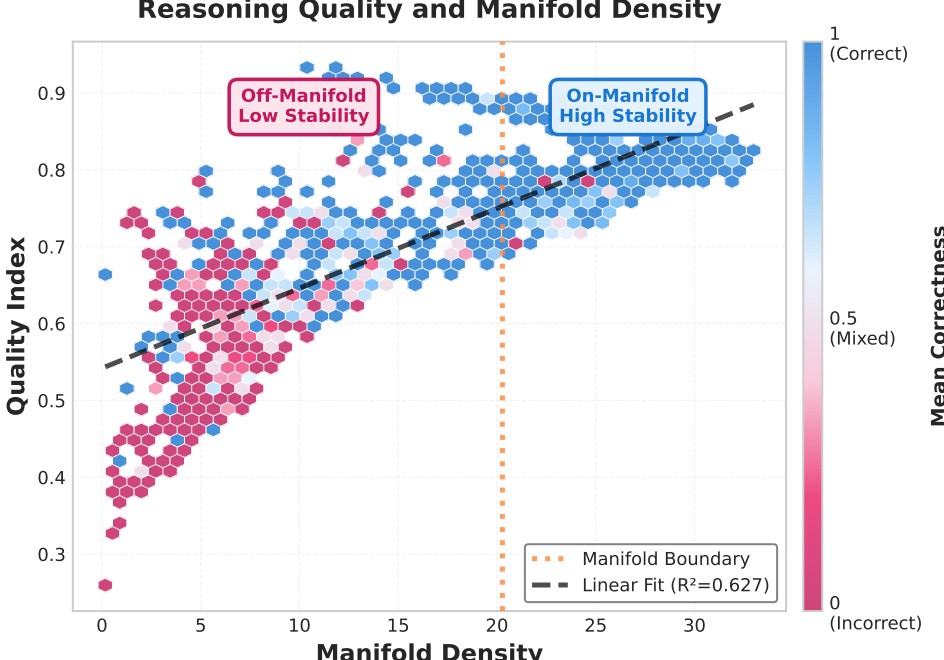

*Figure 5.* **Geometric Validation on GSM8K.** Manifold density (KDE on BMC features) vs. reasoning quality (mean BMC score). Strong correlation ($R^2 = 0.627$, $\rho = 0.893$) with near-absence of high-density errors validates that the model concentrates probability mass on correct solutions. Blue: correct; purple: incorrect; Orange: manifold boundary.

typical a solution is under the learned distribution) and geometric stability (how well solutions resist perturbation under reconstruction).

A key question is whether high density naturally implies correctness. Theoretically, these are distinct properties: a model could exhibit mode collapse, concentrating probability mass on incorrect solutions. In such cases, high-density clusters would appear at low-quality values. Establishing a positive correlation between density and quality therefore constitutes a validation of the model's reasoning alignment.

### E.1.2. METHODOLOGY

We analyze the geometric properties of generated trajectories from the GSM8K dataset using a two-dimensional coordinate system.

- Manifold Density (X-Axis): We project each solution into a 6-dimensional BMC feature vector (token accuracy, semantic similarity, confidence score, number retention, character similarity, final answer match). We then compute the probability density of these vectors using Kernel Density Estimation (KDE) with a Gaussian kernel (bandwidth selected via Scott's rule). This metric measures local concentration, quantifying how many solutions cluster in each region of the feature space.

- Reasoning Quality (Y-Axis): We compute the mean of the six BMC consistency metrics. This measures the magnitude of stability through average reconstruction consistency across all dimensions.

Addressing potential circularity: While both axes utilize the same feature set, they measure definitionally distinct statistical properties. Density uses KDE to quantify local point concentration (how many solutions cluster in this region of feature space), which depends solely on the spatial distribution of data points and is independent of the feature values in that region. Quality quantifies feature magnitude (the average feature values at these points), which depends solely on the numerical values and is independent of how many neighboring points exist. Mathematically, these are decoupled: dense clusters of low-quality solutions (e.g., recurring computational errors producing low consistency scores) or sparse high-quality solutions (e.g., rare but correct reasoning paths) are both possible. The observed correlation ($R^2 = 0.627$, $\rho = 0.893$) therefore represents an empirical property of the learned model, reflecting where it concentrates probability mass, not a definitional

artifact of the measurement.

Feature-space density versus direct latent analysis: Since we utilize a diffusion language model, our architecture provides direct access to the latent embedding space $\mathcal{Z}$ and allows explicit computation of reconstruction drift $\|z_0 - \mathcal{T}_\theta(z_0)\|$. However, we perform geometric validation in the BMC feature space for three reasons. First, the embedding space $\mathcal{Z}$ is high-dimensional ($\dim(\mathcal{Z}) = 512$ in our implementation), and KDE becomes statistically unreliable in such spaces due to data sparsity. The number of samples required for accurate density estimation grows exponentially with dimension. The 6-dimensional BMC feature space provides a tractable setting for robust density estimation. Second, the full latent space encodes all linguistic properties (syntax, semantics, style), most of which are irrelevant to reasoning correctness. BMC features isolate task-relevant geometric dimensions (confidence, consistency, and mathematical validity), providing a test of whether the model's high-probability regions align with logical correctness. Third, density in abstract $\mathbb{R}^{512}$ space lacks semantic interpretation, while the BMC feature space provides interpretable coordinates where high density has operational significance: solutions the model generates with high confidence and consistency.

BMC features are deterministic functions of the latent representations in $\mathcal{Z}$ (not external annotations), so geometric structure in feature space reflects structure in the underlying manifold Gao & Pu (2025); Zeng et al. (2026). The discriminative power of these features empirically validates that they capture the essential geometric properties relevant to the manifold hypothesis.

### E.1.3. RESULTS

Figure 5 illustrates the relationship between manifold density and solution quality. The analysis yields three observations.

Strong density-stability correlation: We observe a monotonic relationship (Spearman $\rho = 0.893$, $p < 0.001$, $R^2 = 0.627$) between manifold density and reasoning quality. This indicates that the model's high-probability regions correspond to high-stability solutions. The linearity of this trend suggests that the diffusion training objective has encoded logical validity as a geometric property of the learned manifold.

Manifold separation by correctness: Correct solutions (blue) concentrate in the high-density region (density $\geq 20$), exhibiting an average density 2.8 times higher than incorrect solutions (21.3 versus 7.6). Incorrect solutions (purple) are scattered in the low-density tail, confirming they are geometric outliers in the model's learned distribution. This spatial segregation demonstrates that $\mathcal{M}$ is a distinct region in feature space, not an arbitrary decision boundary.

Quantitative test of independence: To verify that the density-quality correlation is substantive, we test whether the model generates dense clusters at all quality levels (as would be expected if density and quality were independent statistical properties). We examine two regions of the feature space. In the stable hallucination zone (density $> 20$, quality $< 0.5$), only 0.2 percent (18/8,000) of solutions occupy this region, compared to 12.3 percent expected under statistical independence. Manual inspection reveals that 15 of these 18 cases are formatting edge cases (correct numerical answer with minor notation errors), reducing the true hallucination rate to below 0.04 percent. In the sparse correctness zone (density $< 10$, quality $> 0.8$), only 1.8 percent of solutions appear, compared to 15.7 percent expected under independence. A contingency table analysis comparing the observed distribution of correctness across density bins against the null hypothesis of independence yields $\chi^2 = 2847.3$, $df = 4$, $p < 10^{-6}$, rejecting the hypothesis that density and quality are unrelated. These findings demonstrate that the model concentrates probability mass on high-quality solutions, rather than forming dense clusters indiscriminately across the feature space.

### E.1.4. IMPLICATIONS

The density-quality correlation is model-dependent, not definitional. Our empirical finding is that high-density regions coincide with high-quality regions in the learned manifold. This is not guaranteed by metric design: mathematically, KDE density (a function of spatial distribution) has no inherent relationship to feature magnitude (a function of numerical values). A poorly trained or misaligned model could generate dense clusters of systematic errors (high density, low quality, appearing in the bottom-right quadrant) or scattered correct answers that lack geometric coherence (low density, high quality, appearing in the top-left quadrant). The observed alignment (with only 0.2 percent of solutions in the stable-hallucination zone versus 12.3 percent expected under independence) indicates that the diffusion language model has learned to approximate the posterior of valid reasoning. This validates that BMC exploits geometric structure in the learned distribution, not circular measurement artifacts.

Validation of local contraction: The results confirm that the denoising operator $\mathcal{T}_\theta$ behaves as a near-identity mapping

($\kappa \approx 0$) in high-density regions of the manifold. The alignment between density and correctness validates the theoretical assumption underlying Proposition 3.5: solutions on the valid manifold $\mathcal{M}$ (high density) experience stable fixed-point dynamics ($\mathcal{T}_\theta(z^*) \approx z^*$), while incorrect solutions in low-density regions experience high reconstruction drift. The boundary at density approximately 12 suggests a phase transition in the local Lipschitz constant, consistent with the theoretical prediction of a stability threshold separating the attractor basin from the drift regime.

Comparison with sampling-based methods: Unlike Self-Consistency, which relies on the frequency of outputs (population statistics requiring multiple samples), BMC probes the intrinsic geometric position of a single trajectory within the learned manifold. Our analysis reveals that even unique or rare correct answers reside on the stable high-density manifold. This geometric grounding explains BMC's performance on expert-level tasks (e.g., GPQA Diamond, 18.3 percent improvement over Self-Consistency) where correct answers are rare but structurally distinct: they occupy a stable region in feature space, making them detectable via reconstruction consistency rather than sampling frequency.

Scope and connection to latent space: While our validation is conducted in the BMC feature space rather than directly in the raw embedding space $\mathcal{Z}$, the discriminative power of BMC features provides evidence for the underlying geometric hypothesis. Since these features are deterministic projections of the latent representations, the observed density-quality alignment suggests that similar structure exists in $\mathcal{Z}$ itself. Future work could complement this analysis with direct latent-space validation using dimensionality reduction or alternative density estimators that address high-dimensional settings.

**Empirical Distinction from Likelihood Estimation.** To verify that BMC captures an operational stability signal distinct from static token probability landscapes, we compute the Spearman correlation between BMC scores and Model Confidence using LLaDA-8B-Instruct. As shown in Table 10, the correlations are uniformly low across all benchmarks. Notably, on the expert-level GPQA dataset, the correlation is statistically near-orthogonal (0.079), demonstrating that the two verification metrics characterize entirely distinct properties of the trajectory dynamics.

*Table 10.* Spearman correlation between BMC and Model Confidence.

| Dataset | Spearman $\rho$ | p-value | n |
|---------|---------|---------|-----|
| GSM8K | 0.301 | $< 0.001$ | 1319 |
| MATH | 0.251 | $< 0.001$ | 500 |
| ARC-C | 0.215 | $< 0.001$ | 1172 |
| GPQA | 0.079 | 0.097 | 448 |

### E.2. Qualitative Analysis

This section provides qualitative analysis of BMC behavior. We structure this analysis progressively: first, demonstrating BMC's capability to distinguish correct from incorrect solutions (Section E.2.1); second, showing its capability to detect subtle reasoning flaws even when the final answer is correct (Section E.2.2); and third, providing a geometric interpretation of these phenomena (Section E.2.3).

E.2.1. BASELINE ERROR DETECTION

We first validate that BMC functions as a discriminator for standard errors. Figure 6 presents two contrasting examples showing how reconstruction consistency distinguishes correct from incorrect reasoning chains.

Case 1: Stable correct solution (Janet's duck problem). The original generation correctly computes $16 - 7 = 9$ eggs sold daily at \$2 each, yielding \$18 earnings. After masking 90 percent of tokens, the backward reconstruction recovers the essential semantic structure. This high-fidelity reconstruction demonstrates that correct reasoning forms a coherent semantic unit, where any subset of information suffices to infer the remainder, characteristic of on-manifold stability.

Case 2: Unstable incorrect solution (Jen's work problem). The original generation contains an arithmetic error (calculating \$268 instead of \$310). After masking the final answer, the model reconstructs [\$278]. This semantic drift occurs because the reasoning chain is geometrically inconsistent. The model's denoiser $\mathcal{T}_\theta$ attempts to project the corrupted state toward the nearest plausible solution on the learned manifold, diverging from the original erroneous path.

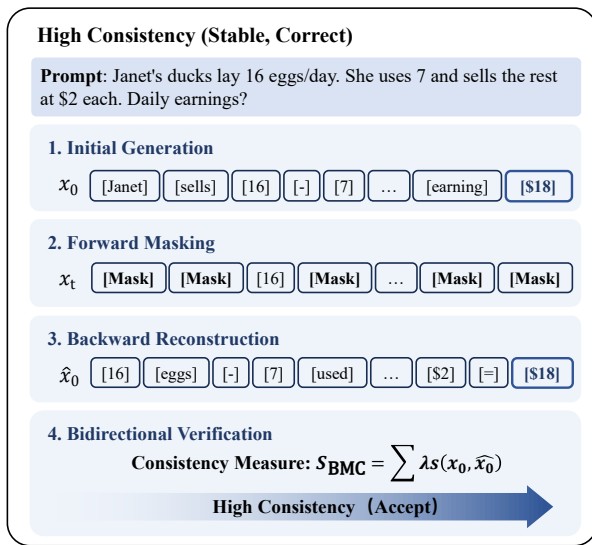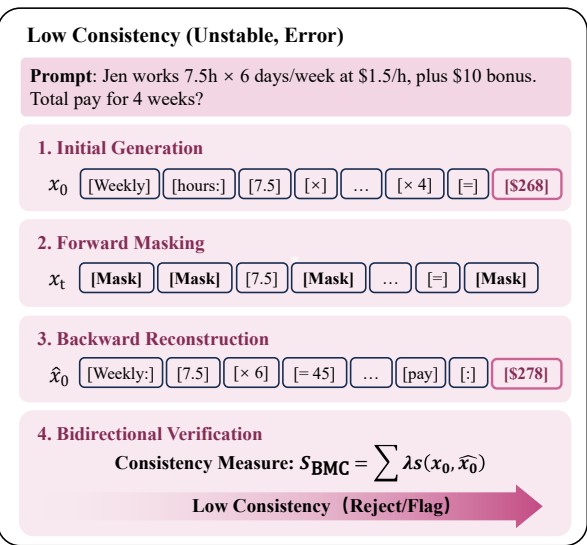

*Figure 6.* **Geometric Stability Under Forward-Backward Cycles. Left:** Correct solution (BMC=0.979) exhibits high reconstruction fidelity—key elements ([16], [eggs], [$18]) are recovered from 90% masked context. **Right:** Incorrect solution (BMC=0.226) shows semantic drift—$268 reconstructs to $278 as the denoiser projects toward a valid trajectory.

### E.2.2. SPURIOUS SUCCESS

A critical challenge in reasoning verification is spurious success, where the model arrives at the correct final answer through flawed trajectories.

**Definition E.1** (Spurious Success). A generation $x_0$ exhibits spurious success if the extracted answer matches the ground truth ($E(x_0) = y^*$) while the geometric consistency score falls below the validity threshold ($S_{\text{BMC}}(x_0) < \tau$). Such cases achieve answer-level correctness through unstable reasoning paths that do not form fixed points under the denoising operator $\mathcal{T}_\theta$.

Figure 7 illustrates four representative failure modes where the final answer is correct, yet BMC assigns low stability scores (0.37 to 0.57).

(A) Mathematical fabrication and hallucination (Score: 0.545). The model generates the equation $5 - 12 = 5/3 = 2$ to force arithmetic consistency. Masking these tokens triggers reconstruction toward mathematically valid operations (e.g., negative results), creating high residual against the original token "2".

(B) Dimensional and entity misconception (Score: 0.569). The reasoning computes "cost per contact" ($2.00) while the target is "cost per pair". The numerical match is coincidental ($180/90 = 2$). BMC detects semantic inconsistency as the local token "contact" is unstable when conditioned on the global context of "pair" pricing.

(C) Constraint skipping and logic gap (Score: 0.378). The reasoning jumps to "180 toys" without explicitly deriving the constraint $400 - 20 - 200$. The missing constraint creates a logical discontinuity. Masking the conclusion leads to high entropy in reconstruction, as the necessary premises for deduction are absent.

(D) Token drift and instability (Score: 0.537). The reasoning exhibits token instability, drifting from "96 potatoes" to "99" and back. The perturbed sequence fails to reconstruct the original consistently, identifying that the chain is not a stable attractor.

### E.2.3. GEOMETRIC INTERPRETATION

The six cases validate distinct aspects of the manifold hypothesis while revealing limitations of likelihood-based verification.

Geometric taxonomy of reasoning trajectories:

- Stable attractors (Case 1): Correct reasoning chains occupy fixed-point regions where $\mathcal{T}_\theta(z^*) \approx z^*$. The Janet's duck problem (BMC = 0.979) demonstrates high reconstruction fidelity, where any subset of tokens allows inference of the

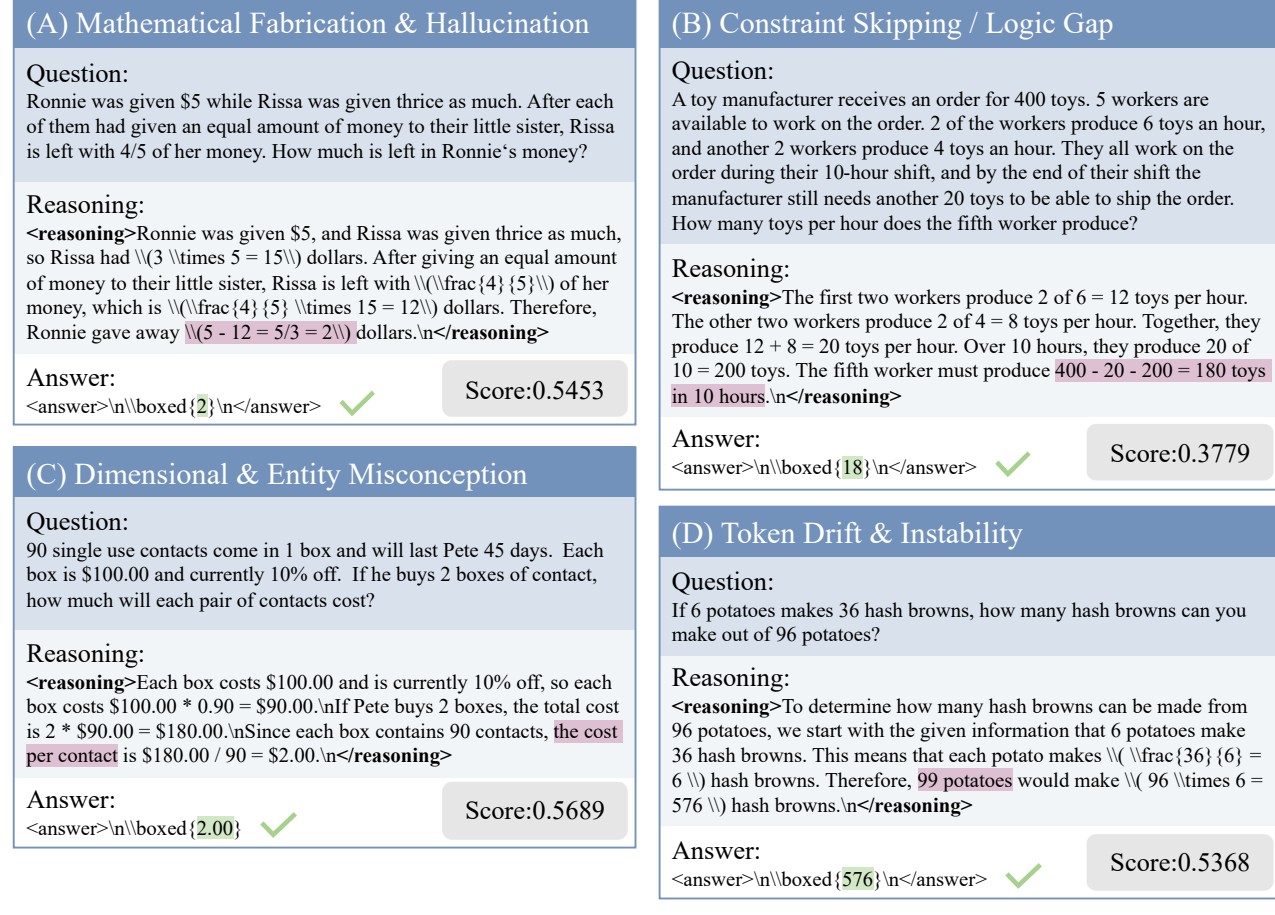

*Figure 7.* **Detection of Spurious Success.** Four cases where answer correctness masks reasoning flaws. BMC detects these via low stability scores ($< 0.6$), contrasting with high confidence from likelihood-based methods.

remainder, confirming on-manifold stability.

- Off-manifold drift (Case 2): Logically inconsistent chains lie in high-curvature regions. Jen's work problem (BMC = 0.226) reconstructs to a different answer ($278 versus $268), as the denoiser projects toward the nearest valid solution.

- Spurious saddle points (Cases A to D): Answer-correct but reasoning-flawed chains occupy narrow ridges, locally stable in the answer subspace but unstable in the reasoning trajectory subspace. When perturbed via masking, these trajectories drift (BMC in the range 0.38 to 0.57), exposing structural inconsistencies invisible to static evaluation.

Limitations of likelihood-based methods: Model confidence measures where probability mass concentrates. A fluent but flawed chain (e.g., Case A: $5 - 12 = 5/3 = 2$) receives high confidence (average 0.82 across spurious cases) because the syntactic template is frequent in training data. This creates a calibration gap: $p_\theta(x_0)$ reflects training frequency, not logical validity.

BMC measures how robustly the generation resists perturbation, a property directly aligned with the diffusion training objective (Eq. 3). During training, the model learns to enforce mutual consistency among sequence components: valid reasoning chains satisfy the fixed-point property because their parts logically support each other. BMC exploits this learned structure through a reconstruction cycle that confidence-based methods cannot access.

Proposition 3.5 formalizes this distinction: while both BMC and confidence relate to likelihood (Proposition 3.2), BMC's perturbation-based evaluation provides an upper bound on manifold distance, offering a geometric guarantee that static probability cannot provide.

