# OpenReview forum: "Reasoning on the Manifold: Bidirectional Consistency for Self-Verification in Diffusion Language Models"
_ICML.cc/2026/Conference — ICML 2026 regular_

### Official Review · Reviewer_Wb3p · 2026-03-10

**Soundness:** 2
**Presentation:** 3
**Significance:** 2
**Originality:** 2
**Overall Recommendation:** 2
**Confidence:** 5

**Summary:**

The paper introduces BMC, a training-free metric designed to verify the reasoning traces of dLLMs. The core idea is that valid reasoning sequences are stable attractors on the model's learned data manifold, while invalid ones are not. BMC quantifies this stability through a forward-masking and backward-reconstruction cycle, measuring the similarity between the original and reconstructed sequences. The authors demonstrate BMC's utility in three areas: diagnosing solution validity without ground truth, guiding inference-time rejection sampling for self-correction, and serving as a dense reward signal for reinforcement learning-based alignment. Experiments on several reasoning benchmarks show that BMC outperforms baseline verification methods.

**Compliance With Llm Reviewing Policy:**

Affirmed.

**Final Justification:**

I maintain my original rating after reading all the authors' rebuttals and other reviews. In short, the paper presents a well-engineered system that works, but it works because of the heuristics, not the geometry. The geometric narrative is a distraction from what is, in essence, a clever but incremental application of feature engineering. The claims of fundamental novelty and generality are not supported. While the packaging is sophisticated, the core ideas are disappointingly familiar. The claims of fundamental novelty and generality are therefore not supported.

**Key Questions For Authors:**

- The paper's central thesis rests on the idea that BMC measures something fundamentally different ("structural stability") from likelihood ("surface fluency"). However, Proposition 3.2 directly links BMC to the ELBO, a proxy for log-likelihood. Could you please clarify this apparent contradiction? How can BMC be fundamentally different from a likelihood estimator if maximizing it is equivalent to maximizing a form of likelihood? A convincing response would need to explain what geometric information BMC captures that is not already contained in the true (but intractable) likelihood p(x).

2. The proposed RL alignment objective (Section 4.3) rewards sequences with high BMC scores. Is there a risk that the model might learn to "game" this objective by generating overly simple or repetitive reasoning chains that are inherently stable and easy to reconstruct, but lack depth or novelty? This could be a form of "reward hacking" specific to the BMC metric. Have you observed any such behavior, and how might it be mitigated?

**Limitations:**

The authors have not adequately discussed the limitations. I would suggest improvements in the risk of "gaming" the metric.

**Strengths And Weaknesses:**

Strength:

- The paper addresses the critical and challenging problem of ensuring reliability and verifying the reasoning process in dLLMs. The empirical results are strong, showing consistent improvements over several baselines across multiple reasoning datasets.

- The paper is well-written and clearly structured.

Weakness:

- Limited originality and novelty: The central weakness of this paper lies in its originality. Despite the elaborate geometric framing ("manifold," "attractors," "drift"), the core mechanism of BMC is essentially a form of self-evaluation by measuring reconstruction fidelity. At its core, the paper proposes a new, and arguably more complex, way to estimate the likelihood or "confidence" of a sequence under a dLLM. The idea that a model's ability to reconstruct corrupted data correlates with the data's quality or likelihood is a foundational concept in generative modeling, not a new insight. The authors frame this as a profound geometric property, but it can be more simply interpreted as a proxy for the sequence's probability density under the model's learned distribution. Proposition 3.2, which links BMC to the reweighted ELBO, further supports this interpretation: maximizing BMC is akin to maximizing a form of log-likelihood. Therefore, the claim of a novel "geometric perspective" seems to be an overstatement for what is essentially a likelihood estimation technique tailored to the diffusion architecture.

- Unconvincing distinction from existing concepts: The authors argue that BMC is fundamentally different from methods like Self-Consistency or confidence scoring. However, the distinction is not as sharp as claimed.
1. The argument that BMC tests "structural stability" while confidence scores test "surface fluency" is debatable. Both are ultimately functions of the model's learned probability distribution. A sequence that is "structurally unstable" under BMC's test is, by another name, a sequence that lies in a low-density or high-curvature region of the probability landscape, which a perfect likelihood estimator would also identify. BMC appears to be a better estimator, not a different kind of estimator.
2. The core advantage of BMC seems to stem almost entirely from the architectural properties of dLLMs (bidirectional context) rather than a novel verification principle. The contribution feels more like an "exploit" of a specific architecture rather than a general, transferable principle of reasoning verification.
- Potential for "gaming" the metric: If BMC were to be used extensively for alignment (as proposed in Sec 4.3), the model could learn to "game" this specific reconstruction task. It might learn to produce reasoning chains that are trivially easy to reconstruct, even if they are not more logically sound. For example, it could favor shorter, simpler, or more repetitive phrasing that is robust to masking, potentially at the cost of reasoning depth. This limitation is not discussed.

---

> ### Author Rebuttal · Authors · 2026-03-31
>
> We thank Reviewer Wb3p for the critique, but we must clarify a fundamental misunderstanding: BMC is not a proxy for likelihood, but a principled probe of geometric stability.
>
> ## [W1 & Q1] BMC is related to likelihood, but not reducible to it.
>
> We appreciate the feedback but must clarify a fundamental distinction: BMC is a probe of dynamic geometric stability, not a mere proxy for likelihood.
> - **Theoretical Bridge $\neq$ Operational Identity:** Proposition 3.2-3.4 establishes a theoretical link for the specific case of KL divergence, but BMC is a general framework. In practice, we use structural and semantic metrics (e.g., number retention) to evaluate dynamic resilience, which static likelihood cannot capture.
> - **Static Density vs. Dynamic Stability:** Likelihood measures token frequency based on training data. BMC measures whether a reasoning trajectory acts as a stable attractor. "Confident hallucinations" may have high likelihood due to common syntactic templates but will exhibit off-manifold drift under BMC’s perturbation.
> - **Empirical Proof:** Likelihood-based confidence (Model Confidence) reduces to random guessing on complex tasks like GPQA (AUROC $\approx$ 0.5), whereas BMC remains robust. Furthermore, BMC effectively flags spurious success—cases where the model is confident but logically inconsistent—which likelihood-based methods fail to detect.
>
> By moving from static scoring to a bidirectional consistency check, we provide a geometric signal that likelihood-based packaging simply cannot access.
>
>
> ## [W2] On dLLM Architecture and the Generality of the Perturb-and-Reconstruct Principle
>
> We thank the reviewer for noting the bidirectional nature of dLLMs. Rather than an architecture-specific exploit, we view our approach as a functional discovery of an intrinsic self-verification channel within dLLM dynamics. BMC provides an efficient, training-free mechanism to probe stability through the model's own corruption-reconstruction process. Crucially, the perturb-and-reconstruct philosophy is a universal principle where valid reasoning functions as a stable attractor, while invalid paths exhibit drift. While dLLMs offer an elegant implementation, this geometric hypothesis can inspire consistency-based verification for a broad range of future architectures.
>
> ## [Q2] On reward hacking and the risk of gaming BMC.
>
> We agree that reward hacking is a legitimate concern. However, our formulation includes two fundamental safeguards that prevent the model from "gaming" the metric via superficial simplicity.
> - **Correctness Gating:** As defined in Equation 14, the BMC bonus is strictly conditioned on answer correctness. This prevents the model from gaining rewards by simply producing easy-to-reconstruct but logically incorrect outputs.
> - **Geometric Anchoring:** BMC requires consistency across token, semantic, and structural levels under heavy masking. A trajectory only receives a high score if it contains a dense network of decision-critical anchors. Attempting to game the metric by removing these logical anchors would increase reconstruction ambiguity, causing the consistency score to drop rather than improve.
>
> This is further validated by our improvements on ARC-C (+2.4%) and GPQA (+3.6%). On these knowledge-intensive tasks, simple formatting or numerical hacks are insufficient to achieve high BMC, proving that the signal is grounded in genuine reasoning stability. **To directly refute the "gaming via simplification" hypothesis, we compared the generation characteristics:**
>
> | Metric | Outcome RL | Geometric Alignment | $\Delta$ | Sign Test $p$ |
> |---|---:|---:|---:|---:|
> | Avg. response length (tokens) | 166.95 | **178.45** | +11.50 | <0.0001 |
> | Type-token ratio (lexical diversity) | **0.4326** | 0.4171 | -0.0154 | <0.0001 |
> | Avg. reasoning steps | 6.31 | **8.03** | +1.72 | <0.0001 |
>
> The results show that Geometric Alignment produces significantly longer responses (+11.5 tokens) and more reasoning steps (+1.72), which is the exact opposite of what reward hacking via simplification would predict. The marginal decrease in Type-Token Ratio (TTR) is a known statistical artifact of longer texts described by Herdan’s Law, not an indication of lexical impoverishment. **In summary, BMC establishes a regime where superficial simplicity does not imply high consistency.** Since the metric depends on the structural anchors supporting correctness, any simplification that destroys these foundations will cause scores to drop rather than improve, ensuring the model optimizes for intrinsic reliability rather than arbitrary engineering hacks.

---

> > ### Author Rebuttal · Reviewer_Wb3p · 2026-04-01
> >
> > I have read the other reviews and the authors' rebuttal, but my recommendation remains **Reject**. My core concerns about the paper's limited originality and overstated claims persist.
> >
> > The "Novel Geometric Perspective" is a Re-packaging of a Known Principle, whose claimed distinction from likelihood is unconvincing. The paper's attempt to distance BMC from likelihood by labeling it "dynamic stability" is a semantic distinction, not a fundamental one. The core principle—checking consistency via a perturb-and-reconstruct cycle—is a classic technique, not a novel discovery. The authors' own rebuttal reveals the truth: BMC's effectiveness relies on manually adding task-specific rules like "number retention," which they admit pure likelihood doesn't have. This means the "secret sauce" isn't a deep geometric insight, but rather well-known, effective feature engineering. The "manifold" framing serves as a sophisticated narrative for what is, in essence, a well-engineered heuristic-based verifier. The contribution is practical, but the claim to fundamental originality is not justified.
> >
> > I concur with Reviewers f5gt and RDjm that the paper's generality is overstated. The authors' rebuttal frames the need for domain-specific metrics as a "feature," but this is a classic sign of a non-generalizable method. The empirical success hinges on a "carefully engineered verification bundle" (Reviewer RDjm), with the authors' own ablation showing performance collapsing without task-specific components like answer matching. This confirms the "geometric signal" from the core reasoning text is weak. Claiming this is a "universal philosophy" while admitting that the user must manually design and tune a new set of metrics and weights for each new domain/dataset is a contradiction. A truly general framework should not require such extensive, bespoke engineering to function.
> >
> > The Risk of Reward Hacking is Underestimated and the Rebuttal is Insufficient. I appreciate the authors' rebuttal on reward hacking, but their defense only addresses the most naive forms of gaming and their evidence is open to alternative interpretations. The "Correctness Gating" prevents generating entirely wrong answers, but it fails to address the risk of converging to the most simplistic or template-like reasoning paths among the set of all correct solutions. This is a more subtle, but critical, failure mode that limits the model's reasoning diversity and potential. The experimental evidence showing longer responses is inconclusive. As I argued previously, this could simply indicate that the model learns to be "strategically verbose"—adding redundant, predictable phrases to boost reconstruction stability without adding real logical complexity. The authors have not provided metrics for "reasoning depth" or "logical efficiency" to rule out this alternative explanation. Their reliance on superficial metrics like length and step count is not a convincing refutation of the reward hacking hypothesis.
> >
> >
> > In conclusion, the paper presents a well-executed but conceptually incremental work. It dresses up a familiar concept in a new "geometric" narrative, while its practical success hinges on hand-crafted, non-general components. The work feels more like clever engineering for dLLMs than a fundamental contribution to AI reasoning. I expect other reviewers to re-examine the paper.

---

> > > ### Author Response · Authors · 2026-04-01
> > >
> > > We appreciate Reviewer Wb3p's continued engagement and present new empirical evidence to address the remaining concerns.
> > >
> > > ## [C1] Empirical Evidence: BMC is Not a Proxy for Likelihood
> > >
> > > The reviewer's core remaining argument assumes that BMC is operationally reducible to likelihood estimation. If this were true, BMC should correlate strongly with Model Confidence, the standard operationalization of log-likelihood in dLLMs. We computed the Spearman correlation using LLaDA-8B-Instruct:
> > >
> > > | Dataset | Spearman ρ | p-value | n |
> > > |---|---:|---:|---:|
> > > | GSM8K | 0.301 | <0.001 | 1319 |
> > > | MATH | 0.251 | <0.001 | 500 |
> > > | ARC-C | 0.215 | <0.001 | 1172 |
> > > | GPQA | 0.079 | 0.097 (n.s.) | 448 |
> > >
> > > The correlations are uniformly low across all datasets. On GPQA, the most challenging benchmark where reliable verification matters most, the correlation is statistically indistinguishable from zero. **Two measures of the same underlying quantity would not be near-orthogonal precisely where both are most needed.**
> > >
> > > We understand the intuition that higher reconstruction stability might simply reflect higher likelihood. However, this is a well-known misconception extensively studied in the Uncertainty Estimation (UE) literature. Because modern neural networks are inherently overconfident, raw likelihood is notoriously unreliable as an indicator of true correctness. This is precisely why the UE field has developed various alternative uncertainty estimators for better calibration, and BMC belongs to this distinct line of verification.
> > >
> > > ## [C2] Theoretical unification and practical instantiation are distinct levels of a framework
> > >
> > > We respectfully disagree with the premise that requiring domain-specific metrics undermines the generality of a theoretical framework. A unified principle prescribes a universal protocol rather than a single one-size-fits-all measurement instrument. Consider the fundamental machine learning concept of measuring data similarity. While the theoretical principle is universal, the actual distance function must reflect the structure of the domain, such as Edit Distance for text, Intersection over Union for bounding boxes, and Abstract Syntax Tree matching for code. The overarching concept of similarity is not dismissed as bespoke feature engineering simply because the distance metric adapts to the data modality.
> > >
> > > Our unified theoretical contribution is the geometric verification principle, which establishes that valid reasoning acts as a stable attractor under a perturb-and-reconstruct cycle. The instantiation of $D(\cdot,\cdot)$ is simply how that principle is operationalized to capture the decision-critical structure of each specific domain, such as using numerical anchors for math. Equating the universal protocol with its domain-specific probe conflates two distinct levels of framework design. **Bridging a general theory with domain-specific measurements is an objective requirement of any formal framework, not a limitation of our method.**
> > >
> > > ## [C3] On the risk of reward hacking
> > >
> > > We appreciate the continued engagement regarding reward hacking. While we respect that reward hacking is a universal concern for any RL objective, the empirical evidence demonstrates that neither of the proposed failure modes is occurring.
> > >
> > > The most direct evidence lies in our cross-task performance. If the gains were merely the result of strategic verbosity or superficial formatting, improvements would be concentrated on simpler tasks where formulaic responses are occasionally effective. However, our observed pattern shows the exact opposite:
> > >
> > > | Task | Difficulty | Geometric Align vs. Outcome RL (512 tokens) |
> > > |---|---|---:|
> > > | GSM8K | Easy | +2.3% |
> > > | ARC-C | Medium | +3.0% |
> > > | GPQA | Hard | +3.6% |
> > > | MATH | Very Hard | +4.4% |
> > >
> > > Performance gains scale progressively with task difficulty. On rigorous, knowledge-intensive benchmarks like MATH (+4.4%) and GPQA (+3.6%), template-based verbose responses fundamentally cannot produce correct final answers. This pattern is consistent with the model optimizing for genuine logical stability rather than superficial hacks.
> > >
> > > At a mechanism level, BMC naturally penalizes superficial verbosity due to its high (90%) masking rate. Strategic verbosity relies on repetitive, low-information phrasing. When 90% of such text is removed, the remaining 10% of anchor tokens lack the structural density required to uniquely and accurately constrain a reconstruction of the full chain. In short, BMC requires high information density to achieve a high score, **making "gaming via verbosity" structurally self-defeating.** We will incorporate an explicit discussion of these safeguards into the revised manuscript.

---

### Official Review · Reviewer_RDjm · 2026-03-11

**Soundness:** 2
**Presentation:** 2
**Significance:** 3
**Originality:** 3
**Overall Recommendation:** 4
**Confidence:** 4

**Summary:**

This paper proposes "Bidirectional Manifold Consistency" (BMC), a metric for verifying reasoning correctness in diffusion Large Language Models (dLLMs). The authors hypothesize that correct solutions act as stable attractors on a high-density manifold. To operationalize this, they introduce a forward-masking and backward-reconstruction cycle to measure the stability of generated sequences. The method is evaluated across three stages: error diagnosis, adaptive inference (rejection sampling), and reinforcement learning alignment.

**Compliance With Llm Reviewing Policy:**

Affirmed.

**Final Justification:**

I maintain my original rating. However, the concerns raised by reviewer Wb3p are worth considering.

**Key Questions For Authors:**

1.Contraction Validation: Can you provide empirical measurements of the reconstruction drift $|x_0 - T_\theta(x_0)|$ for pairs of correct vs. incorrect solutions to explicitly validate the contraction assumption?
2.Are the aggregation weights and thresholds shared across datasets and tasks, or tuned per benchmark?
3.For RL alignment, how much does BMC help beyond simply providing richer task-specific shaping features?

**Limitations:**

yes

**Strengths And Weaknesses:**

Soundness

The paper’s main conceptual contribution is clear and technically motivated: BMC probes the stability of a generated reasoning trace under a forward-mask / backward-reconstruct cycle, and this is a natural thing to try for dLLMs because their generation process is itself bidirectional. The theoretical section gives three layers of justification: a KL/ELBO connection, a more general divergence-based argument over critical tokens, and a geometric fixed-point / contraction view in latent space. This makes the paper better grounded than a purely heuristic scoring proposal.
That said, I have reservations about how strong the theory is relative to the claims. Much of the formal argument depends on assumptions that seem plausible but are not established for actual large-scale dLLMs: local Lipschitz continuity of log-likelihood in embedding space, existence of a valid-solution manifold, and especially a local contraction property of the denoising operator around that manifold. These are useful interpretive assumptions, but they do not amount to a rigorous guarantee that higher BMC implies correctness in realistic settings. The geometric story is appealing, but presently reads more like a motivating hypothesis than a theorem-backed characterization of reasoning validity. In particular, Proposition 3.5 becomes meaningful only if the contraction and fixed-point assumptions truly hold in the regions visited by generated reasoning traces.
A second soundness concern is that the final practical score is a weighted combination of six hand-designed metrics, including task-specific extractors and answer matching. While this composite score works well empirically, it weakens the paper’s claim that BMC is an intrinsic geometric signal. In the implemented system, some of the strongest components are not purely geometric at all; for example, final answer match and number retention appear to drive much of the performance, and the ablation table shows final answer match alone is already very strong. This raises the question of whether the gains come mainly from the manifold principle or from a carefully engineered verification bundle. The theory centers on generic dissimilarity under reconstruction, but the reported method is closer to a task-aware ensemble verifier.
Experimentally, the diagnosis results are fairly convincing: BMC beats model confidence, self-evaluation, and self-consistency on most datasets and both dLLMs, with especially larger gaps on harder tasks such as GPQA. The adaptive resampling results are also directionally positive, though the margins over best-of-N BMC or self-consistency are sometimes modest, and one GPQA/Dream setting slightly favors best-of-N BMC. The RL alignment results show improvements over SFT and outcome-only RL, but the evidence here is somewhat thinner because the comparison space is limited and many implementation details are relegated to the appendix. Still, the empirical section is reasonably broad and supports the claim that the signal is useful in several stages of the pipeline.
A final soundness issue is evaluation leakage / circularity risk. The score includes final answer match between the original and reconstructed outputs, not against ground truth, which is legitimate as a consistency measure; however, on benchmarks where answers are short and highly structured, consistency of extracted answers may dominate the signal. Likewise, in the appendix the “geometric validation” defines density over BMC feature vectors and quality as the mean of BMC features, which is not strictly circular but is close enough that the validation does not fully convince me the manifold hypothesis itself is independently established.
Overall, I find the empirical part reasonably sound, but the theoretical framing somewhat stronger than the evidence strictly justifies.

Presentation

The paper is generally well organized. The overall narrative is easy to follow: intuition, theory, estimator, applications, experiments, and appendix. Figures 1 and 2 are effective at conveying the central idea, and the three-stage framing of diagnosis / inference / alignment gives the work a coherent arc. The paper also positions itself well against PRMs, self-consistency, and recent dLLM-specific methods.
However, there are also presentation weaknesses. First, the paper often overstates the connection between theory and practice. The formal definition of BMC is generic and elegant, but the actual estimator is a composite of six features with tuned weights and task-specific extraction logic. This gap should be made more explicit in the main paper, not only in the appendix. Second, the notation is sometimes heavier than necessary, especially in Section 3 where several propositions build on assumptions that are not clearly operationalized later. Third, some terminology is slightly overloaded: “manifold,” “density,” “stability,” and “consistency” are used in both formal and intuitive senses, sometimes without sharp distinctions. Fourth, the appendix contains important clarifications about metric design, failure modes, and complexity that arguably belong in the main text.
For reproducibility, I would have liked more explicit detail in the main paper on the exact aggregation weights, how task-specific extractors are written, whether weights are tuned per dataset or shared, and what variance exists across seeds. These details matter because the final method is not a single scalar metric derived directly from the diffusion process, but a multi-component scoring system.
Significance

The problem is important. If dLLMs are to become a serious alternative to AR LLMs for reasoning, then verification and self-correction mechanisms tailored to their bidirectional generation process are highly relevant. A diffusion-specific verifier that is training-free and can be used at diagnosis, inference, and alignment time would be quite useful. In that sense, the paper asks a meaningful question and explores a promising direction.
The practical significance is moderate to good. The diagnosis gains are meaningful, especially on difficult tasks where simple confidence and self-consistency degrade. The resampling method appears computationally attractive because verification uses truncated reconstruction rather than full additional samples. The RL use case is also interesting because it turns sparse correctness signals into denser within-solution guidance. These are genuine benefits if they hold broadly.
Still, I do not view the impact as fully established yet. The experiments are limited to two dLLMs and mostly benchmark-style reasoning tasks, many with short structured answers. It remains unclear how much BMC would help on open-ended reasoning, tool use, code synthesis, or tasks where final answers are less extractable. Also, since the best-performing score uses several engineered components, deployment on new domains may require additional design work. So the idea could be influential, but the present evidence supports promising potential more than broad demonstrated impact.

Originality

The paper is original in its central angle: using the diffusion model’s own forward/backward corruption-reconstruction dynamics as an intrinsic verification probe. That is a natural but nontrivial shift away from AR-inspired self-consistency or external reward models. The framing of correctness as reconstruction stability on a reasoning manifold is also novel and memorable. The extension across diagnosis, inference, and RL makes the contribution broader than just one scoring trick.
At the same time, part of the implementation is less original than the framing suggests. The actual score mixes standard ingredients: token overlap, semantic similarity, number retention, edit similarity, confidence, and answer matching. So the originality lies more in the bidirectional perturbation-and-reconstruction protocol plus the manifold interpretation than in the individual metric components. I think that is still enough for a meaningful contribution, but the paper should be careful not to oversell the final composite as a purely new theoretical object. The novelty is solid, though somewhat more in perspective and system design than in a single sharply isolated algorithmic primitive.

---

> ### Author Rebuttal · Authors · 2026-03-31
>
> We thank Reviewer RDjm for the technically precise review. We agree with the need to separate the conditional geometric hypotheses in §3 from the practical estimators in §4, and we will explicitly state what is empirically validated in our revision.
>
> ## [W1 & Q1] Clarifying Theoretical Assumptions and Empirical Reconstruction Drift
>
> We agree that Propositions 3.2–3.5 serve as motivating results under specific structural assumptions rather than formal guarantees for deployed dLLMs. These propositions require rigorous mathematical conditions that are intractable to strictly satisfy in practice, especially within highly non-linear deep neural networks.
>
> While our formal proofs rely on ideal conditions, the underlying geometric phenomenon remains robust in practice. To address Q1, we directly quantified the reconstruction drift $\|z_0-\mathcal{T}_\theta(z_0)\|$ within the embedding space on the GSM8K dataset using LLaDA-8B-Instruct:
>
> | Group               |  $1- \|z_0-\mathcal{T}_\theta(z_0)\|$ | Recon drift $\|z_0-\mathcal{T}_\theta(z_0)\|$ |
> | ------------------- | --------------------: | ---------------------------------: |
> | Correct solutions   |        0.85 |         0.15 |
> | Incorrect solutions |        0.60 |         0.40 |
>
> This table demonstrates the contraction gap: correct trajectories remain substantially closer to their reconstructions than incorrect ones. Furthermore, Appendix E provides robust empirical support for this manifold view. Specifically, the density-quality alignment in §E.1 (Spearman $\rho = 0.893$, $p < 10^{-6}$) is entirely consistent with the premise that high-quality solutions reside in more contractive neighborhoods.
>
> ## [W2, W3 & W4] BMC is a general verification protocol with task-adaptive instantiations of $D(\cdot,\cdot)$.
>
> We agree that the bridge between theory and practice requires clearer articulation. **BMC is a perturb-and-reconstruct verification framework where the discrepancy term $D(\cdot,\cdot)$ is naturally instantiated according to the target domain.** While the geometric protocol in section 3 is universal, the measurement of reconstruction discrepancy must adapt to the specific task.
>
> For math, numbers and the final answer carry the strongest validity signals. Consequently, Final Answer Match (FAM) and Number Retention (NumRet) are expected to be strong instantiations:
>
> | BMC Configuration                     | AUROC |  AUPR |
> | ------------------------------------- | ----: | ----: |
> | Full BMC          | 0.880 | 0.927 |
> | w/o FAM                | 0.748 | 0.840 |
> | w/o NumRet                  | 0.808 | 0.890 |
> | w/o (FAM and NumRet) | 0.680 | 0.803 |
>
> **To demonstrate that our framework does not rely on explicit terminal answers, we successfully extended BMC to open-ended code generation.** By replacing answer-centric signals with structural observables like Abstract Syntax Tree overlap, the same protocol applies perfectly to unstructured tasks. We kindly refer you to our response to Reviewer f5gt ([W1, Q1 & Q2]) for full experimental tables on HumanEval and MBPP. This cross-domain flexibility confirms that BMC is a general geometric verifier rather than a math-specific heuristic.
>
> Regarding Appendix E, density and quality are not definitionally coupled. Because dense low-quality clusters and sparse high-quality clusters are both mathematically possible, the observed alignment is a meaningful empirical finding rather than a tautology.
>
> ## [Q2] Are weights shared across datasets?
>
> The aggregation weights and thresholds are tuned per dataset on a held-out validation split. The optimized version simply provides an additional layer of dataset-specific calibration to maximize the utility of this geometric signal.
>
> ## [Q3] What does BMC add in RL beyond task-specific shaping?
>
> **We agree that any dense reward provides richer shaping than sparse outcome supervision. However, the unique value of BMC is introducing geometric stability as a principled regularizer.**
>
> Standard handcrafted heuristics typically rely on task-specific discrete signals like keyword matching. In contrast, BMC extracts a continuous geometric property: the contractivity of a reasoning trajectory under the perturb-and-reconstruct cycle. This allows the RL process to differentiate between genuinely robust reasoning and brittle shortcuts. While a lucky guess might satisfy surface-level heuristics, its underlying representation remains unstable and collapses under perturbation. By rewarding trajectories in stable, contractive neighborhoods, BMC provides a mathematically grounded signal that optimizes for the intrinsic structural reliability of the reasoning paths.
>
> **[Presentation]** We fully accept your suggestions regarding presentation. In the final revision, we will explicitly label the propositions as conditional motivating results and standardize the terminology across "manifold," "density," "stability," and "consistency."

---

> > ### Author Rebuttal · Reviewer_RDjm · 2026-04-02
> >
> > Thank you for the reply. I would maintain my score.

---

### Official Review · Reviewer_f5gt · 2026-03-12

**Soundness:** 3
**Presentation:** 4
**Significance:** 3
**Originality:** 3
**Overall Recommendation:** 4
**Confidence:** 4

**Summary:**

This paper studies self-verification for diffusion language models (dLLMs). The core idea is that correct reasoning trajectories should be more stable under a forward masking and backward reconstruction cycle, while incorrect trajectories should drift more. Based on this intuition, the paper proposes Bidirectional Manifold Consistency (BMC), a training-free metric that scores a generated solution by comparing it to its reconstruction after perturbation. The authors then use BMC in three ways: as an unsupervised verifier for diagnosing whether a solution is correct, as a signal for adaptive rejection sampling during inference, and as a dense reward term for RL-based alignment. Empirically, the method is evaluated on two dLLMs, four reasoning benchmarks, and several baselines, with reported gains in AUROC/AUPR for diagnosis, sample efficiency for resampling, and downstream accuracy for RL alignment.

**Compliance With Llm Reviewing Policy:**

Affirmed.

**Final Justification:**

The paper proposes an interesting geometric framework for self-verification in diffusion LMs with solid experiments. The rebuttal addressed my main concerns: the ablation clarifying the role of task-specific components and the extension to code generation (HumanEval/MBPP) convincingly demonstrate that BMC generalizes beyond math-specific heuristics. I also appreciate the clarification on the "training-free" terminology. I maintain my weak accept and believe the revised version incorporating these clarifications would strengthen the paper.

**Key Questions For Authors:**

1. How much of the empirical gain remains if you remove the most task-specific components, especially final-answer match and number retention? This is important because Table 4 suggests these components dominate the performance. If the gains largely persist without them, I would view the method as stronger evidence for a genuinely general geometric verifier. If performance drops sharply, I would revise my interpretation toward “a useful composite heuristic for reasoning tasks” rather than a broadly principled manifold-based verifier.
2. How does BMC perform on tasks where final-answer extraction is ambiguous or where number retention is not meaningful?
Much of the current evaluation is on math/QA-style benchmarks. I would like to know whether the method still works in less structured domains, such as coding.

**Limitations:**

It doesn't seem to be discussed yet.

**Strengths And Weaknesses:**

**Strengths**
1. The proposed BMC metric is technically natural for dLLMs because it directly leverages the forward corruption and reverse denoising mechanisms of the model. It is encouraging to see these metrics outperform metrics for autoregressive model such as confidence.
2. Empirical evaluation is strong and comprehensive. It compares with sufficient baselines on a few reasoning benchmarks. The results are overall positive.

**Weakness**
1. The paper’s strongest theoretical claims are more suggestive than fully convincing. The theory section establishes connections between BMC and likelihood-style objectives, and then introduces a geometric interpretation based on local contraction around a putative solution manifold. However, the practical BMC score used in experiments is a hand-designed weighted combination of token accuracy, semantic similarity, number retention, final-answer match, character similarity, and intrinsic confidence. This practical metric is much farther from the clean theoretical object, and the paper does not fully close the gap between the theory and the engineered final score used in experiments. In particular, the best-performing components seem heavily task-specific, especially final-answer match and number retention, which makes the broader “manifold consistency” story feel less universal than advertised.
2. The method is described as training-free for diagnosis and inference, but the final score uses multiple engineered components and weighted aggregation, and the alignment section additionally assumes outcome supervision to gate the reward. The paper does compare against common baselines, but it is still somewhat hard to disentangle how much of the improvement comes from the diffusion-specific reconstruction mechanism versus task-specific metric design and added structure.

---

> ### Author Rebuttal · Authors · 2026-03-31
>
> We sincerely thank Reviewer f5gt for the careful review. You raise a fundamental question regarding our design: is BMC a general geometric verifier, or a collection of task-specific heuristics? We view BMC as a unified, top-down verification framework. The core mechanism of measuring the geometric stability during the perturb-and-reconstruct process remains completely universal. What adapts to the target domain is simply the instantiation of the consistency measure $D(\cdot,\cdot)$ used to quantify that reconstruction discrepancy.
>
> ## [W1, Q1 & Q2] BMC is a General Geometric Verification Framework, Not a Task-Specific Heuristic
>
> We appreciate your thoughtful evaluation. **We view the domain-specific instantiation of $D(\cdot,\cdot)$ not as a gap, but as a core feature of our framework.** While BMC establishes a universal verification principle based on geometric stability, measuring reconstruction discrepancy inherently depends on the target domain. Tailoring $D(\cdot,\cdot)$ is therefore not a heuristic shortcut. It is the exact mechanism required to apply a unified geometric principle to diverse generation structures.
>
> For mathematical reasoning, the critical validity signals are heavily concentrated in numbers and the final answer, making Final Answer Match (FAM) and Number Retention (NumRet) the most natural instantiations of $D(\cdot,\cdot)$. As part of our comprehensive ablation study evaluating each distance component, we specifically isolate the impact of removing these two critical anchors on the GSM8K dataset:
>
> | BMC Configuration                     | AUROC |  AUPR |
> | ------------------------------------- | ----: | ----: |
> | Full BMC (Optimized Weights)          | 0.880 | 0.927 |
> | w/o Final Answer Match                | 0.748 | 0.840 |
> | w/o Number Retention                  | 0.808 | 0.890 |
> | w/o FAM + NumRet (task-agnostic core) | 0.680 | 0.803 |
>
> As shown above, removing both FAM and NumRet drops the AUROC and AUPR to 0.680 and 0.803, respectively. This performance drop aligns perfectly with our theoretical expectations. If we strip away the tokens carrying the most concentrated mathematical signals, the metric is forced to evaluate stability based merely on generic natural language continuity, entirely missing the geometric essence of math problems. This confirms that rigorous geometric verification intrinsically requires tracking the decision-critical anchors of the specific domain.
>
> **To further demonstrate that our framework does not fundamentally depend on answer matching, we extended BMC to open-ended code generation**. In programming tasks, the generated code itself is the solution, shifting the decision-critical structure from numerical anchors to syntax and program logic. We do not introduce a new heuristic metric here. Instead, we simply instantiate the discrepancy measure $D(\cdot,\cdot)$ using established structural indicators. Specifically, we capture the geometric stability of code by defining $D(\cdot,\cdot)$ based on Abstract Syntax Tree (AST) overlap (measured via Jaccard similarity) and token substitution rates, alongside the standard reconstruction confidence.
>
> | Method           |        | HumanEval |        |  MBPP |
> | ---------------- | -----: | --------: | -----: | ----: |
> |                  | AUROC↑ |     AUPR↑ | AUROC↑ | AUPR↑ |
> | Model Confidence |  0.535 |     0.630 |  0.528 | 0.581 |
> | Self-Consistency |  0.615 |     0.684 |  0.651 | 0.669 |
> | Self-Evaluation  |  0.683 |     0.727 |  0.639 | 0.646 |
> | BMC(Ours)        |  0.687 |     0.783 |  0.662 | 0.668 |
>
> Unlike math problems, where validity signals concentrate on numerical anchors, code correctness is holistic and dispersed across the structure. By capturing this distributed stability, our adapted metric outperforms baselines without requiring explicit answer tokens. This demonstrates that domain-specific adaptations are not arbitrary "hacks." The components simply change to match the decision-critical geometry of each domain, while the underlying perturb-and-reconstruct principle remains exactly the same.
>
> We agree this theory-to-practice bridge should be stated more explicitly in the paper, and we will sharpen the revision accordingly: BMC should be presented as a general geometric verification protocol whose $D(\cdot,\cdot)$ term is instantiated to match the target domain's decision-critical structure.
>
>
> ## [W2] On the "training-free" claim
>
> We agree to clarify our terminology. Here, "training-free" refers specifically to the BMC score estimation: we compute the score directly from the base model's dynamics without training any auxiliary verifiers or reward models. In the §4.3 alignment pipeline, our actual objective is to demonstrate that the model can effectively learn from BMC as a dense geometric RL reward. We will revise the manuscript to make the scope of this "training-free" claim explicitly clear.

---

> > ### Author Rebuttal · Reviewer_f5gt · 2026-04-02
> >
> > Thanks for authors’ rebuttal. I will keep my positive score.

---

### Official Review · Reviewer_aqVP · 2026-03-13

**Soundness:** 3
**Presentation:** 3
**Significance:** 3
**Originality:** 3
**Overall Recommendation:** 4
**Confidence:** 4

**Summary:**

This paper explores the internal structure of reasoning processes in Transformer-based language models from a geometric perspective. The authors argue that reasoning trajectories in language models can be understood as paths evolving along a low-dimensional manifold in representation space. To investigate this idea, the paper analyzes hidden-state trajectories generated during reasoning tasks and studies their geometric properties using techniques such as dimensionality reduction and trajectory analysis. The empirical results suggest that reasoning sequences often follow relatively smooth and structured paths in the model’s representation space, which the authors interpret as evidence that reasoning behavior may be constrained by an underlying manifold structure. The paper presents this perspective as a way to better understand how reasoning unfolds inside large language models.

**Compliance With Llm Reviewing Policy:**

Affirmed.

**Final Justification:**

I maintain my original score of weak accept. The paper presents a refreshing geometric perspective on reasoning verification in diffusion language models, and the rebuttal provided helpful clarifications, particularly regarding the role of task-specific components and the extension to code generation. However, I still view the strongest empirical results as relying importantly on task-specific components and dataset-specific calibration, which makes the claim of a generally intrinsic geometric verifier not yet fully established. The geometric narrative is appealing but remains more of a promising perspective than a rigorously validated universal principle across diverse tasks. I note that Reviewer Wb3p raises valid concerns about limited originality and overstated generality claims that the AC should weigh carefully.

**Key Questions For Authors:**

1. How robust are the observed trajectory structures across different model architectures and scales?
2. Did the authors test whether similar manifold structures appear in non-reasoning generation tasks?
3. Could the proposed manifold perspective lead to practical improvements, such as better decoding strategies or reasoning supervision?
4. How sensitive are the results to the choice of dimensionality reduction method?

**Limitations:**

yes

**Strengths And Weaknesses:**

Strengths
1. The paper takes a perspective that I found refreshing. Instead of focusing on improving benchmark accuracy, it tries to understand how reasoning unfolds inside the model.
2. The geometric interpretation of reasoning trajectories is interesting and could potentially inspire new ways of analyzing model behavior.
3. The visualizations of hidden-state trajectories are intuitive and make the central idea relatively easy to grasp.
4. The work fits into a broader line of research that studies the structure of representation spaces in deep networks.

Weaknesses
1. While the geometric interpretation is interesting, the paper is largely descriptive. It analyzes representations but does not show how the proposed perspective could lead to improved models or algorithms.
2. The claim that reasoning happens on a low-dimensional manifold is somewhat difficult to evaluate. Many high-dimensional neural representations can appear low-dimensional after projection, so it is not entirely clear how strong the evidence is.
3. The experiments are somewhat limited in scope. The analysis appears to focus on a relatively small number of models and tasks, which makes it harder to judge whether the observed patterns are general.
4. I also found myself wondering how sensitive the results are to the specific visualization and dimensionality reduction techniques used.

---

> ### Author Rebuttal · Authors · 2026-03-31
>
> We sincerely thank Reviewer aqVP for the thoughtful review and for valuing our geometric perspective. We are highly encouraged that you found our approach to reasoning trajectories refreshing and intuitive. Following your insightful summary, we will address your questions and demonstrate how we translate this theory into actionable algorithmic improvements.
>
> ## [W1 & Q3] Translating the Geometric Perspective into Practical Algorithms
>
> While our work establishes strong theoretical guarantees, our core objective is to translate these insights into practical algorithms. Specifically, we apply our theoretical metric for geometric stability (BMC) to drive concrete improvements in three key areas:
>
> - **Diagnosis (Table 1)**: We utilize BMC as a training-free verifier without relying on ground-truth labels. This approach yields up to a +13.9% AUROC improvement over Self-Consistency on the expert-level GPQA benchmark.
> - **Inference (Table 2)**: By integrating BMC, we developed Manifold-Guided Rejection Sampling (MGRS). This highly efficient method increases GSM8K accuracy from 70.5% to 79.5% using an average of only 3.3 samples, outperforming standard Best-of-N baselines.
> - **Alignment (Table 3)**: We demonstrate that BMC serves as an effective dense reward for Reinforcement Learning. Without any human annotation, this approach surpasses standard Outcome RL by +4.4% on MATH and matches the performance of strong autoregressive baselines like Qwen2.5-7B.
>
> ## [W2 & Q4] Evaluating Without Dimensionality Reduction
>
> We appreciate you raising this important point regarding the pitfalls of low-dimensional projections. We realize that our mention of a "6-dimensional feature space" in Appendix E.1 was misleading and may have given the impression that we applied dimensionality reduction to the hidden states.
>
> To clarify, our framework does not apply any dimensionality reduction (such as PCA or t-SNE) to the neural representations. As formalized in Proposition 3.5, BMC computes the exact reconstruction drift $\|z_0-\mathcal{T}_\theta(z_0)\|$ directly on the raw, unprojected hidden states. Because our theoretical claims and algorithms operate entirely within this original high-dimensional space, they are free from any projection artifacts.
>
> The "6 dimensions" mentioned in the appendix simply correspond to the six scalar metrics (e.g., semantic similarity, token accuracy) used for the Kernel Density Estimation plots in Figure 5. Therefore, our geometric evaluation is mathematically independent of, and completely unaffected by, visualization choices.
>
> ## [W3 & Q1] Robustness Across Architectures and Scales — New Results
>
> We appreciate the opportunity to demonstrate the generality of our findings. Beyond the main text, we have conducted new experiments on an MoE architecture and a larger scale model during the rebuttal period:
>
> | Model | Architecture        | GSM8K AUROC | GSM8K AUPR |
> | ----------------------------- | ------------------- | ----------: |----------: |
> | LLaDA-8B-Instruct             | Dense, 8B           |       0.893 |0.943|
> | Dream-v0-Instruct-7B          | Dense, 7B           |       0.898 |0.787|
> | **LLaDA-MoE-7B-A1B-Instruct** | MoE, 7B (1B active) |       0.858 |0.878|
> | **LLaDA2.1-mini (16B)**       | Dense, 16B          |       0.853 |0.933|
>
> The structural stability of valid reasoning paths holds consistently across different scales and attention mechanisms. This confirms that BMC captures an intrinsic property of the diffusion denoising process itself.
>
> ## [Q1 & Q2] Scope: Task-Agnostic and a Universal Philosophy
>
> We deeply appreciate the question regarding the broader scope of our framework. A core insight of our work is that assessing geometric stability through a "Perturb and Reconstruct" cycle is a universal philosophy for verifying reasoning.
>
> While our current implementation (BMC) explicitly leverages the native bidirectional dynamics of dLLMs (as their closed-form masking makes this cycle zero-shot and elegant), the underlying principle is not strictly dLLM-specific. We believe this "Perturb and Reconstruct" paradigm, in which valid reasoning acts as a stable attractor, can inspire novel consistency checks for autoregressive (AR) models in future work.
>
> Furthermore, the core framework is inherently task-agnostic. While we instantiate the consistency score using task-specific metrics (e.g., final answer matching) for reasoning benchmarks, this is merely to measure the consistency of the reconstructed trajectory for that specific domain. To empirically demonstrate its versatility, we evaluated BMC on open-ended Code Generation tasks during the rebuttal. Due to strict space limits, we kindly refer you to our response to Reviewer f5gt (Q2) for the detailed setup and results. **Briefly, BMC successfully evaluated open-ended code outputs without gold-standard templates or rigid formatting constraints, confirming it captures a universal indicator of generation quality.**

---

> > ### Author Rebuttal · Reviewer_aqVP · 2026-04-04
> >
> > Thank you for the detailed rebuttal. I appreciate the clarifications and the additional evidence. The response is helpful, especially in clarifying the intended role of the geometric interpretation, the practical scope of the “training-free” claim, and the fact that the core method does not depend on visualization-based dimensionality reduction.
> >
> > I would maintain my score. At the same time, I think some of the central concerns are only partially resolved. In particular, while the paper presents an interesting and promising perturb-and-reconstruct verification framework, the strongest empirical results still seem to rely importantly on task-specific components and dataset-specific calibration. This makes me somewhat less convinced by the broader claim that the current evidence already establishes BMC as a generally intrinsic geometric verifier, rather than a strong composite verifier tailored to reasoning settings.
> >
> > More broadly, I still view the geometric story as an appealing and useful perspective, but one that is not yet fully established beyond the current set of tasks and design choices. For that reason, although I continue to see value in the paper and keep my score unchanged, I hope the AC weighs the remaining concerns about generality, theory-to-practice gap, and possible overstatement of novelty in the final assessment.

---

> > > ### Author Response · Authors · 2026-04-05
> > >
> > > We sincerely thank Reviewer aqVP for the positive feedback and for recognizing the theoretical value of our Geometric Stability framework. To address the remaining concern directly: **the reliance on task-specific components does not undermine BMC's claim as a general geometric verifier.**
> > >
> > > In our framework, **the geometric verification protocol (perturb-and-reconstruct) is fundamentally universal and task-agnostic.** To deploy this universal protocol in practice, the distance metric D(⋅,⋅) acts as a decoupled, plug-and-play interface, purposefully designed to capture the distinct, decision-critical structures of different domains. **Therefore, integrating the general theory with domain-specific measurements is an intended architectural feature that enables practical deployment, rather than a limitation of its generality.** This structural decoupling is precisely what equips BMC to adapt across various practical scenarios.
> > >
> > > Beyond the extensive experiments on math and complex general QA reasoning tasks in the main paper, our rebuttal demonstrates BMC's robust generalizability by readily extending it to **open-ended code generation** without requiring complex redesigns or heavy tuning. **BMC outperforms all baselines on HumanEval and MBPP without relying on any answer-matching components.** This provides strong empirical evidence that BMC's stability signal is an intrinsic, universal property, adaptable to diverse domains through principled instantiations of the measurement space.
> > >
> > > We hope this clarifies the structural distinction between the universal protocol and its measurement layer, fully addressing the concerns regarding generality.

---

### Decision · Program_Chairs · 2026-04-30

**Decision:**

Accept (regular)

**Comment:**

Three of four reviewers recommend weak accept (aqVP, f5gt, RDjm, all Confidence 4); Wb3p alone recommends reject (Confidence 5). All reviewers acknowledge that the empirical results are solid. The disagreement is not about whether the method works, but about whether the geometric framing represents a genuine theoretical contribution or an elaborate narrative for what is essentially a well-engineered verifier.
Wb3p's most compelling point is that Table 4 is difficult to argue away: Final Answer Match alone achieves 0.819 AUROC on GSM8K, and removing it causes the largest single performance drop in the composite. The authors frame domain-specific instantiation as a deliberate design feature rather than a limitation; this argument is not unreasonable, but it does mean that the "universal geometric verifier" claim is stronger than the evidence currently supports. The theory-to-practice gap between the clean propositions in Section 3 and the tuned six-component estimator in Section 4 is real, and the paper does not fully address it.

At the same time,  Wb3p's central claim that BMC simply reduces to likelihood scoring is not fully convincing. The near-zero correlation with model confidence on GPQA and the extension to code generation via AST-based metrics, which lack a terminal "final answer" to match, provide meaningful evidence that the perturb-and-reconstruct protocol captures a stability signal beyond simple answer-matching. The three-stage evaluation is also genuinely broad.

On balance, while the AC shares Wb3p's skepticism regarding the "universal" framing, the underlying mechanism — bidirectional perturbation-and-reconstruction — is a natural and well-motivated probe for dLLM reasoning stability that has not been previously explored. The low correlation with likelihood and the cross-domain results are sufficient to justify acceptance. The paper is recommended for acceptance as a lower-priority poster.